# FedAvg Converges to Zero Training Loss Linearly for Overparameterized Multi-Layer Neural Networks

## Abstract

Federated Learning (FL) is a distributed learning paradigm that allows multiple clients to learn a joint model by utilizing privately held data at each client. Significant research efforts have been devoted to develop advanced algorithms that deal with the situation where the data at individual clients have heterogeneous distributions. In this work, we show that data heterogeneity can be dealt from a different perspective. That is, by utilizing a certain overparameterized multi-layer neural network at each client, even the vanilla FedAvg (a.k.a. the Local SGD) algorithm can accurately optimize the training problem: When each client has a neural network with one wide layer of size $N$ (where $N$ is the number of total training samples), followed by layers of smaller widths, FedAvg converges linearly to a solution that achieves (almost) zero training loss, without requiring any assumptions on the clients' data distributions. To our knowledge, this is the first work that demonstrates such resilience to data heterogeneity for FedAvg when trained on multi-layer neural networks. Our experiments also confirm that, neural networks of large size can achieve better and more stable performance for FL problems.

## 1 Introduction

In Federated Learning (FL), multiple clients collaborate with the help of a server to learn a joint model McMahan et al. (2017). The privacy guarantees of FL has made it a popular distributed learning paradigm, as each client holds a private data set and aims to learn a global model without leaking its data to other nodes or the server. The performance of FL algorithms is known to degrade when training data at individual nodes originates from different distributions, referred to as the *heterogeneous data setting* Yu et al. (2019a); Woodworth et al. (2020a). In the past few years, a substantial research effort has been devoted towards developing a large number of algorithms that can better deal with data heterogeneity, Karimireddy et al. (2020b); Zhang et al. (2021); Li et al. (2018); Acar et al. (2020); Khanduri et al. (2021). However, in practice it has been observed by a number of recent works, that in spite of the data heterogeneity, the simple vanilla FedAvg algorithm (a.k.a. the Local SGD) still offers competitive performance in comparison to the state-of-the-art. For example, see Table 2 in Karimireddy et al. (2020a), Table 1 in Reddi et al. (2020), and Table 2 in Yang et al. (2021) for performance comparison of FedAvg on popular FL tasks.

Motivated by these observations, we ask: Is it possible to handle the the data heterogeneity issue from a different perspective, without modifying the vanilla FedAvg algorithm? To answer this question, in this work we show that FedAvg can indeed perform very well *regardless* of the heterogeneity conditions, if the models to be learned are nice enough. Specifically, FedAvg finds solutions that achieve almost zero training loss (or almost global optimal solution) very quickly (i.e., linearly), when the FL model to be trained is certain overparameterized multi-layer neural network. To the best of our knowledge, this is the first result that shows (linear) convergence of FedAvg in the overparameterized regime for training multilayer neural networks. The major contributions of our work are listed below.

- Under certain assumptions on the neural network architecture, we prove some key properties of the clients' (stochastic) gradients during the training phase (Lemmas 1 and 2). These results allow us to establish convergence of FedAvg for training overparameterized neural networks without imposing restrictive heterogeneity assumptions on the gradients of the local loss functions.

- We design a special initialization strategy for training the network using FedAvg. The initialization is designed such that the singular values of the model parameters and the outputs of the first layer of local and aggregated model parameters stay positive definite during the training. This property combined with overparameterization enables FedAvg to converge linearly to a (near) optimal solution.
- We conduct experiments on CIFAR-10 and MNIST datasets in both i.i.d. and heterogeneous data settings to compare the performance of FedAvg on various network architectures of different sizes.

To our knowledge, this is the first work that shows the linear convergence of FedAvg (both SGD and GD versions) to the optimal solution when training a overparameterized multi-layer neural networks.

**Related Work: Federated Learning (FL).** FL algorithms were first proposed in McMahan et al. (2017), where within each communication round the clients utilize their private data to update the model parameters using multiple SGD steps. Earlier works analyzed the performance of FedAvg for the case of homogeneous data setting Zhou and Cong (2018); Stich (2018); Lin et al. (2020); Woodworth et al. (2020b); Wang and Joshi (2021), i.e., when the local data at each client follows the same underlying distribution. Motivated by practical applications, recent works have analyzed FedAvg for heterogeneous client data distributions Yu et al. (2019b;a); Haddadpour and Mahdavi (2019); Woodworth et al. (2020a) and it was observed that the performance of FedAvg degrades as the data heterogeneity increases. To address the data heterogeneity issue among clients, many works have focused on developing sophisticated algorithms Karimireddy et al. (2020b); Zhang et al. (2021); Acar et al. (2020); Li et al. (2018); Khanduri et al. (2021); Karimireddy et al. (2020a); Das et al. (2020).

**Overparameterized Neural Networks.** The surprising performance of overparameterized neural networks[1] has raised significant research interest in the ML community to analyze the phenomenon of overparameterization Belkin et al. (2019). Consequently, many works have analyzed the performance of centralized (stochastic) gradient descent (S)GD on overparameterized neural network architectures under different settings Jacot et al. (2018); Li and Liang (2018); Arora et al. (2019); Du et al. (2018; 2019); Allen-Zhu et al. (2019); Zou and Gu (2019); Nguyen and Mondelli (2020); Nguyen (2021).

However, there are only a handful of works that have attempted to analyze the performance of overparameterized neural networks in the distributed setting Li et al. (2021); Huang et al. (2021); Deng and Mahdavi (2021). The works most closely related to our work are Huang et al. (2021) and Deng and Mahdavi (2021). Huang et al. (2021) analyzed the performance of FedAvg on a single hidden-layer neural network for the case when each client utilizes GD for the local updates. The authors established linear convergence of FedAvg using the NTK parameterization and showed that it suffices to design the neural network of width $\Omega(N^4)$ to achieve this performance (where $N$ is the number of training samples). Similarly, Deng and Mahdavi (2021) analyzed the performance of FedAvg on a ReLU neural network but when each client utilizes SGD (or GD) for the local updates. The authors proved convergence of FedAvg under the standard parameterization while requiring the very large network width of $\Omega(N^{18})$. Note that since individual clients can be devices with limited computational capabilities, in realistic settings it is undesirable to have networks of such large widths. In contrast to both these works, we focus on the more practical setting of a multi-layer neural network Nguyen and Mondelli (2020) and establish linear convergence of FedAvg even for the case when each client utilizes SGD for the local updates. Importantly, we show that with proper initialization, it only requires a network of width $N$ at each client, which is much smaller compared to the unrealistic requirements of Huang et al. (2021); Deng and Mahdavi (2021).

## 2 PROBLEM SETUP

In this section, we define the multi-layer neural network and formalize the problem we aim to solve. We consider a distributed system of $K$ clients with each client having access to a privately held data set. We assume that each client $k \in \{1, \ldots, K\}$ has $N_k$ training samples denoted as $\{(X_k, Y_k)\}$, with $X_k \in \mathbb{R}^{N_k \times d_{\text{in}}}$ and $Y_k \in \mathbb{R}^{N_k \times d_{\text{out}}}$. Note that each row of $X_k$ and $Y_k$ represents the feature vector and its corresponding label, and $d_{\text{in}}$ and $d_{\text{out}}$ denote the feature (input) and label (output) dimensions, respectively. We further denote $N = \sum_{k=1}^{K} N_k$ as the total samples across all clients.

---

[1]A model is generally referred to as overparameterized if the number of (trainable) parameters are more than the number of training samples $N$.

Suppose each client trains a fully-connected neural network with $L$ layers, and with activation function $\sigma : \mathbb{R} \to \mathbb{R}$. We denote the vectorized parameters at each node $k \in \{1, \ldots, K\}$ as $\theta_k = [\text{vec}(W_{1,k}), \ldots, \text{vec}(W_{L,k})] \in \mathbb{R}^D$, where $W_{l,k} \in \mathbb{R}^{n_{l-1} \times n_l}$ represents the weight matrix of each layer $l \in \{1, \ldots, L\}$ and $n_l$ represents the width of each layer. Note that each layer inputs a (feature) vector of dimension $n_{l-1}$ and outputs a (feature) vector of dimension $n_l$. For simplicity, define $n_0 = d_{\text{in}}$ and $n_L = d_{\text{out}}$ as the input and the output dimensions of the neural network. We define $F_{l,k}$ as the local output of each layer $l$ at client $k$, then using the above notations, we have

$$F_{l,k} = \begin{cases} X_k & l = 0 \\ \sigma(F_{l-1,k} W_{l,k}) & l \in \{1, 2, \ldots, L\} \\ F_{L-1,k} W_{L,k} & l = L \end{cases}. \tag{1}$$

We further define the vectorized output of each layer and the labels at each client as $f_{l,k} = \text{vec}(F_{l,k}) \in \mathbb{R}^{N_k n_l}$ and $y_k = \text{vec}(Y_k) \in \mathbb{R}^{N_k n_L}$.

Similar to the above setup, we also define the notations to describe a single network, with the full data $(X, Y)$ with $X \in \mathbb{R}^{N \times d_{\text{in}}}$ and $Y \in \mathbb{R}^{N \times d_{\text{out}}}$ as input. This "centralized" network will be useful later to perform the analysis. Then given parameter $\theta = [\text{vec}(W_1), \ldots, \text{vec}(W_L)]$, the output at each layer of the network is defined as

$$F_l = \begin{cases} X & l = 0 \\ \sigma(F_{l-1} W_l) & l \in \{1, 2, \ldots, L\} \\ F_{L-1} W_L & l = L \end{cases}. \tag{2}$$

Next, we define the local and global loss functions. First, each client $k \in \{1, \ldots, K\}$ has a local loss function given by: $\Phi_k(\theta) := \frac{1}{2N_k} \|f_{L,k}(\theta) - y_k\|_2^2$, where $\|\cdot\|_2$ denotes the standard $\ell_2$-norm. Then the global loss function is the sum of weighted local loss functions, given by:

$$\Phi(\theta) := \sum_{k=1}^{K} \frac{N_k}{N} \Phi_k(\theta) = \frac{1}{2N} \|F_L(\theta) - y\|_F^2. \tag{3}$$

Additionally, define the gradient of (3) as $g := [\text{vec}(\nabla_{W_1} \Phi(\theta)), \ldots, \text{vec}(\nabla_{W_L} \Phi(\theta))]$, which is the stacked gradient of the loss w.r.t. the 1st to $L^{\text{th}}$ layer's parameters; define the gradient of the losses at each client $k \in [K]$ as: $g_k := [g_{1,k}, \ldots, g_{L,K}]$ with $g_{l,k} := \text{vec}(\nabla_{W_{l,k}} \Phi(\theta))$ for all $l \in [L]$.

Next, we define the optimality criteria to solve (3) using an overparameterized neural network.

**Definition 1** ($\epsilon$-optimal solution). *Consider an overparameterized problem $\min_\theta \Phi(\theta)$, where there exist $\theta^*$ such that $\Phi(\theta^*) = 0$. A solution $\theta$ is called an $\epsilon$-optimal solution if it satisfies $\Phi(\theta) \leq \epsilon$. Moreover, if $\theta$ is a random variable, then we use $E[\Phi(\theta)] \leq \epsilon$ to denote an $\epsilon$-optimal solution, where the expectation is taken w.r.t. the randomness of $x$.*

## 3 THE FEDAVG ALGORITHM

A classical algorithm to solve problem (3) is the FedAvg McMahan et al. (2017). In FedAvg, each client performs multiple local updates before sharing their updated parameters with the server. We refer the algorithm as FedAvg-SGD (resp. FedAvg-GD) if the clients employ SGD (resp. GD) for the local updates.

The detailed steps to implement FedAvg-SGD are listed in Algorithm 1. We execute the algorithm for a total of $T$ communication rounds, within each communication round every client performs $r$ local updates. In each communication round $t$ the server aggregates the local parameters and constructs $\bar{\theta}^{rt}$ from each client's local parameters $\theta_k^{rt+r}$ and shares it with the clients. The clients use the aggregated parameter, $\bar{\theta}_k^{rt+r}$, as the initial parameter value for computing the next round of local updates. For each $v \in \{0, 1, \ldots, r-1\}$, to update the local parameters the clients compute the (unbiased) stochastic gradient using $m$-samples drawn form their private data set $(X_k, Y_k)$. We denote the random sample drawn at $v^{\text{th}}$ local step in the $t^{\text{th}}$ communication round as $(\tilde{X}_k^{rt+v}, \tilde{Y}_k^{rt+v})$. Using the stochastic gradient estimate, the clients update their parameters locally by employing the SGD step. After $r$ local SGD steps, each client shares its updated parameters with the server and gets back the aggregated parameters before starting the next round of updates. Note that if we choose the batch size $m = N_k$, for all $k \in \{1, \ldots, K\}$, FedAvg-SGD becomes FedAvg-GD.

---

**Algorithm 1** The FedAvg-SGD Algorithm

---

**Initialize:** Parameters $\theta_k^0 = \theta^0$, Step-size $\eta$, # of communication rounds, local updates $T$, $r$
**for** $t = 0, 1, \ldots, T - 1$ **do**
    **for** each client $k \in \{1, \ldots, K\}$ **do**
        Set $\theta_k^{rt} = \bar{\theta}^{rt}$
        **for** $v = 0, 1, \ldots, r - 1$ **do**
            Sample mini-batch of size $m$, $(\tilde{X}_k^{rt+v}, \tilde{Y}_k^{rt+v})$
            Compute stochastic gradient $\tilde{g}_k^{rt+v}$ using (5)
            Update: $\theta_k^{rt+v+1} = \theta_k^{rt+v} - \eta\tilde{g}_k^{rt+v}$.
    Aggregation: $\bar{\theta}^{r(t+1)} = \sum\limits_{k=1}^{K} \frac{N_k}{N}\theta_k^{rt+r}$
**Return:** Parameters, $\bar{\theta}^{rT}$

---

Algorithm 1 summarizes the above description. For each communication round $t \in \{0, 1, \ldots, T - 1\}$ and local step $v \in \{0, 1, \ldots, r-1\}$, we define the vector $\theta_k^{rt+v} := [\text{vec}(W_{1,k}^{rt+v}), \ldots, \text{vec}(W_{L,k}^{rt+v})]$. For FedAvg-SGD, define $\tilde{F}_{l,k}^{rt+v}$ and $\tilde{f}_{l,k}^{rt+v}$ as the output and the vectorized output of each hidden layer $l$, respectively, when the input to the client $k$'s local network is the stochastic (mini-batch) samples $(\tilde{X}_k^{rt+v}, \tilde{Y}_k^{rt+v})$. Using the notation $\tilde{y}_k^{rt+v} = \text{vec}(\tilde{Y}_k^{rt+v})$ as the vectorized labels of the stochastic samples at each local step, we define the mini-batch stochastic loss as:

$$\tilde{\Phi}_k(\theta_k^{rt+v}) := \frac{1}{2m}\|\tilde{f}_{L,k}^{rt+v} - \tilde{y}_k^{rt+v}\|_2^2, \quad (4)$$

and the stochastic gradient as $\tilde{g}_k^{rt+v} := [\tilde{g}_{1,k}^{rt+v}, \ldots, \tilde{g}_{L,k}^{rt+v}]$, where $\tilde{g}_{l,k}^{rt+v}$ is the stochastic gradient w.r.t. the $l^{\text{th}}$ layer of the network evaluated at the $k^{\text{th}}$ client:

$$\tilde{g}_{l,k}^{rt+v} := \text{vec}\left(\nabla_{W_{l,k}}\tilde{\Phi}_k(\theta_k^{rt+v})\right) \in \mathbb{R}^{n_{l-1}n_l}. \quad (5)$$

For each communication round, let us define the aggregated parameters as:

$$\bar{\theta}^{rt} := \left[\text{vec}(\bar{W}_1^{rt}), \cdots, \text{vec}(\bar{W}_L^{rt})\right], \quad \bar{W}_l^{rt} = \sum_{k=1}^{K} \frac{N_k}{N} W_{l,k}^{rt}. \quad (6)$$

For FedAvg-GD, we denote $g_k^{rt+v} := [g_{1,k}^{rt+v}, \ldots, g_{L,k}^{rt+v}]$ as the full gradient of $k^{\text{th}}$ client's loss function, where similar to (5) $g_{1,k}^{rt+v}$ defines the gradient of the loss function w.r.t. the $l^{\text{th}}$ layer's parameters. Throughout, we make the following standard assumption Ghadimi and Lan (2013).

**Assumption 1.** *The stochastic gradients at each client are unbiased, i.e., we have* $\mathbb{E}[\tilde{g}_k^{rt+v}] = g_k^{rt+v}$ $\forall k \in [K]$.

Next, we analyze the performance of the FedAvg for an overparameterized neural network.

## 4 CONVERGENCE ANALYSIS

We present the convergence guarantees of FedAvg when training an overparameterized neural network. We first present a set of assumptions on the network architecture, and activation functions.

**Assumption 2.** *The width of each hidden layer satisfies:* $n_1 \geq N$, $n_2 \geq n_3 \geq \ldots \geq n_L \geq 1$.

**Assumption 3.** *The activation function $\sigma(\cdot)$ in (1) satisfies the following:* 1) $\sigma'(x) \in [\gamma, 1]$; 2) $|\sigma(x)| \leq |x|$; $\forall x \in \mathbb{R}$; 3) $\sigma'$ is $\beta$-Lipschitz, with $\gamma \in (0, 1)$ and $\beta > 0$.

**Remark 1.** *Assumptions 2 and 3 play an important role in our analysis. They help ensure that the local and global loss functions and their (stochastic) gradients are well behaved. Note that Assumption 2 only requires the first layer to be wide while the rest of the layers can be of constant width. Assumption 2 is required to establish a PL like property for the global and local loss functions Nguyen and Hein (2018); Nguyen and Mondelli (2020). Assumption 3 is also standard in the analysis of overparameterized neural networks. Similar assumptions on the smoothness of the activation functions have been made in the past Jacot et al. (2018); Du et al. (2019); Nguyen and Mondelli (2020); Huang and Yau (2020) and are utilized to manage the behavior of the gradients of the loss functions. Importantly, note that as demonstrated in Nguyen and Mondelli (2020) activation functions satisfying Assumption 3 can be utilized to uniformly approximate the ReLU function to arbitrary accuracy.*

**Remark 2.** *We do not impose any assumptions on the distribution of individual clients' local data sets. In contrast, a majority of works on FL impose restrictive assumptions on the gradients (and/or the Hessians) of each client's local loss functions to guarantee algorithm convergence Yu et al. (2019b); Li et al. (2018); Yu et al. (2019a); Karimireddy et al. (2020a). Below, we list two most*

*popular heterogeneity assumptions (from Yu et al. (2019a) and Koloskova et al. (2020), respectively):*

$$\|\nabla\Phi_k(\theta) - \nabla\Phi(\theta)\| \le \delta, \ \forall\theta, \in \mathbb{R}^D, \ \forall\, k \in [K], \text{ for some } \delta > 0. \tag{7}$$

$$\frac{1}{K}\sum_{k=1}^{K}\|\nabla\Phi_k(\theta)\| \le \delta_1 + \delta_2\|\nabla\Phi(\theta)\|, \ \forall\, \theta \in \mathbb{R}^D, \text{ for some } \delta_1, \delta_2 > 0. \tag{8}$$

*Both conditions impose strong restrictions on the gradients of the local clients, and they do not hold for even simple quadratic loss Khaled et al. (2019); Zhang et al. (2021). We will see shortly that, our results will indicate that as long as the neural network is large enough, then the local (stochastic) gradients will be well-behaved, thereby eliminating the need to impose any additional assumptions on the data distributions.*

In the following, we show the convergence guarantees achieved by FedAvg. Our analysis roughly follows the four steps presented below:

**[Step 1]** We first show a key result, that the ratio of the local stochastic gradients and the local full gradients stays bounded (Lemma 1). This result is crucial for the FedAvg-SGD analysis, as it allows us to work with the full local gradients directly, and it helps to bound the gradient drift across local updates within each communication round.

**[Step 2]** Using the result of Step 1, we bound the summation of (stochastic) gradients and the gradient drift during the local updates within each communication round (Lemma 2). This result ensures that irrespective of the data heterogeneity, the gradients size will not change too much from their initial values at the beginning of each round.

**[Step 3]** We then show that adopted network architecture allows us to derive bounds on the size of the gradients and ensure the loss function to be PL during the each communication round (Lemma 3). Utilizing this and the results derived in Steps 1 and 2, we show that the expected loss (3) converges linearly to zero (Proposition 1).

**[Step 4]** Finally, we find a special initialization strategy so that all the conditions imposed on the network properties are satisfied during the entire training process.

Next, let us begin with Step 1. We need the following definition.

**Definition 2.** *Given parameter $\theta_k^{rt+v}$, we define the following quantity for each $k \in [K]$, $t \in \{0, 1, \ldots, T-1\}$ and $v \in \{0, 1, \ldots, r-1\}$: $\rho(\theta_k^{rt+v}) := \|\tilde{g}_k^{rt+v}\|_2 / \|g_k^{rt+v}\|_2$.*

Clearly, $\rho(\theta_k^{rt+v})$ measures the ratio of the norm of stochastic and full gradients of the local loss functions. In the following, we show that if the model parameters at each client satisfy certain conditions, then $\rho(\theta_k^{rt+v})$ is uniformly bounded. Define $\sigma_{\max}(\cdot)$ and $\sigma_{\min}(\cdot)$ as the largest and smallest singular value of a matrix, respectively.

**Lemma 1.** *Let Assumptions 2 and 3 hold. Suppose in any iteration $rt + v$, $v \in \{0, 1, \cdots, r-1\}$, for $\theta_k^{rt+v} = [\text{vec}(W_{1,k}^{rt+v}), \ldots, \text{vec}(W_{L,k}^{rt+v})]$, there exists constant $\bar{\Lambda}_l, \underline{\Lambda}_l, \underline{\Lambda}_F > 0$ such that the singular values of $W_{l,k}^{rt+v}$ and $F_{1,k}^{rt+v}$ satisfy*

$$\begin{cases} \sigma_{\max}(W_{l,k}^{rt+v}) \le \bar{\Lambda}_l, \ l \in [L], \ k \in [K], \\ \sigma_{\min}(W_{l,k}^{rt+v}) \ge \underline{\Lambda}_l, \ l \in \{3, \ldots, L\}, \ k \in [K], \\ \sigma_{\min}(F_{1,k}^{rt+v}) \ge \underline{\Lambda}_F, \ k \in [K]. \end{cases} \tag{9}$$

*where $\lambda_{i \to j} := \prod_{l=i}^{j} \lambda_l$ for given layer-wise parameter $\lambda_l$, then: $\rho(\theta_k^{rt+v}) \le \dfrac{LN\frac{\bar{\Lambda}_{1 \to L}}{\min_{l \in [L]} \bar{\Lambda}_l}}{m\gamma^{L-2}\underline{\Lambda}_{3 \to L}\underline{\Lambda}_F}.$*

As discussed earlier in Step 1, this lemma is crucial to our analysis as it allows us to work with full gradients of individual clients. Before proceeding to Step 2, we need the following definitions:

$$\bar{g}^{rt+v} := \sum_{k=1}^{K}\frac{N_k}{N}g_k^{rt+v} \ \text{ and } \ \bar{\tilde{g}}^{rt+v} := \sum_{k=1}^{K}\frac{N_k}{N}\tilde{g}_k^{rt+v}.$$

Here $\bar{g}^{rt+v}$ and $\bar{\tilde{g}}^{rt+v}$ are the weighted averages of the full and stochastic gradients, respectively. Next, in Step 2 (Lemma 2) we first bound the size of the sum of $\bar{\tilde{g}}^{rt+v}$ over the local updates within each communication round. Then we bound the change in $\bar{g}^{rt+v}$ from $v = 0$ to any $v \in \{0, 1, \ldots, r-1\}$. Note that this quantity measures the drift in the averaged gradients from the start of each communication round.

**Lemma 2.** *For FedAvg-SGD, given step size $\eta > 0$, $v \in \{0, 1, \ldots, r-1\}$ and $q \in \{0, 1, \ldots, v-1\}$. Suppose there exists constants $\bar{\Lambda}_l$, $\rho$, and $A > 0$ such that the following conditions hold:*

$$\bar{\Lambda}_l \geq \sup_{k \in [K]} \sigma_{\max}\left(W_{l,k}^{rt+q}\right), \quad \rho \geq \sup_{k \in [K]} \rho\left(\theta_k^{rt+q}\right), \quad \Phi_k(\theta^{rt+q}) \leq A^q \cdot \Phi_k(\bar{\theta}^{rt}), \ k \in [K].$$

*Then we have*

$$\left\|\sum_{q=0}^{v} \bar{\tilde{g}}^{rt+q}\right\|_2 \leq \frac{\rho L \|X\|_F}{N} \frac{A^{\frac{v+1}{2}} - 1}{\sqrt{A} - 1} \frac{\bar{\Lambda}_{1 \to L}}{\min_{l \in [L]} \bar{\Lambda}_l} \|f_L(\bar{\theta}^{rt}) - y\|_2. \tag{10}$$

*Further, for all $k \in [K]$, $\exists Q_k > 0$, such that we have*

$$\left\|\bar{g}^{rt+v} - \bar{g}^{rt}\right\|_2 \leq \frac{\eta \rho L}{N} \frac{\bar{\Lambda}_{1 \to L}}{\min_{l \in [L]} \bar{\Lambda}_l} \frac{A^{\frac{v+1}{2}} - 1}{\sqrt{A} - 1} \sqrt{\sum_{k=1}^{K} Q_k^2 \|X_k\|_F^2} \|f_L(\bar{\theta}^{rt}) - y\|_2. \tag{11}$$

Next, we show Step 3, that the averaged parameter $\bar{\theta}^{rt}$ defined in (6), after $t$th communication round, will have good performance. Towards this end, we define the full gradient given parameter $\bar{\theta}^{rt}$ as

$$g^{rt} := [\text{vec}(\nabla_{W_1} \Phi(\bar{\theta}^{rt})), \ldots, \text{vec}(\nabla_{W_L} \Phi(\bar{\theta}^{rt}))]. \tag{12}$$

**Lemma 3.** *Let Assumptions 2 and 3 hold. At each communication round $rt$, suppose there exists constant $\bar{\Omega}_l, \underline{\Omega}_l, \underline{\Omega}_F$, such that*

$$\begin{cases} \sigma_{\max}(\bar{W}_l^{rt}) \leq \bar{\Omega}_l, \ l \in [L], \\ \sigma_{\min}(\bar{W}_l^{rt}) \geq \underline{\Omega}_l, \ l \in \{3, \ldots, L\}, \\ \sigma_{\min}(F_1(\bar{\theta}^{rt})) \geq \underline{\Omega}_F, \end{cases} \tag{13}$$

*where $\bar{\theta}^{rt}$ and $\bar{W}_l^{rt}$ are defined in (6). Then we have*

$$\|g(\bar{\theta}^{rt})\|_2 \geq \| \text{vec}\left(\nabla_{W_2} \Phi\left(\bar{\theta}^{rt}\right)\right)\|_2 \geq \frac{\gamma^{L-1}}{N} \underline{\Omega}_{3 \to L} \underline{\Omega}_F \left\|f_L(\bar{\theta}^{rt}) - y\right\|_2, \tag{14}$$

$$\|g(\bar{\theta}^{rt})\|_2 \leq \frac{L}{N} \frac{\bar{\Omega}_{1 \to L}}{\min_{l \in [L]} \bar{\Omega}_l} \left\|f_L(\bar{\theta}^{rt}) - y\right\|_2. \tag{15}$$

**Remark 3.** *Note that (14) is a PL-type inequality Karimi et al. (2016), and requires the special structure of the network that satisfies Assumption 2 Nguyen and Hein (2018); Nguyen and Mondelli (2020). Also, (15) can be proven using Assumption 3.*

Now, we utilize the results of Steps 1 - 2 and Lemma 3 to derive the convergence of FedAvg.

**Proposition 1.** *Use Algorithm 1 to minimize (3). Suppose Assumptions 1, 2 and 3 are satisfied, and for each iteration $rt + v$, $v \in \{0, 1, \cdots, r-1\}$, $\theta_k^{rt+v}$ satisfies the conditions in Lemmas 1 and 2; and for each communication round $rt$, $\bar{\theta}^{rt}$ satisfies conditions in Lemma 3, then $\exists \eta > 0$ such that*

$$E[\Phi\left(\bar{\theta}^{rt}\right)] \leq \left(1 - \frac{r\eta}{N} \gamma^{2(L-2)} \underline{\Omega}_{3 \to L}^2 \underline{\Omega}_F^2\right)^t \Phi\left(\theta^0\right). \tag{16}$$

**Remark 4.** *Proposition 1 above shows that, if the conditions in Lemmas 1, 2 and 3 are satisfied, i.e., we have well-behaved gradients (Lemmas 1 and 2) and PL condition (Lemma 3), we achieve linear convergence of expected loss function for solving (3) with FedAvg-SGD.*

We outline the major steps in the proof of Proposition 1.
*Proof Sketch.* Consider the $t^{th}$ communication round, and suppose the singular values of the parameters satisfy (13), then it is easy to show that $\Phi(\bar{\theta}^{rt})$ is Lipschitz smooth with some constant $Q > 0$. Then using the Lipschitz smoothness of $\Phi(\bar{\theta}^{rt})$, we get

$$\Phi(\bar{\theta}^{r(t+1)}) \leq \Phi(\bar{\theta}^{rt}) - \eta \left\langle g^{rt}, \bar{\tilde{g}}^{rt} + \ldots + \tilde{\tilde{g}}^{rt+r-1}\right\rangle + \frac{Q}{2} \eta^2 \left\|\bar{\tilde{g}}^{rt} + \ldots + \bar{\tilde{g}}^{rt+r-1}\right\|_2^2.$$

Taking expectation on both sides and conditioning on $\bar{\theta}^{rt}$ and the past, we get the following

$$E[\Phi(\bar{\theta}^{r(t+1)})] \leq E\Big[\Phi(\bar{\theta}^{rt}) - \eta\langle g^{rt}, \bar{g}^{rt} + \ldots + \bar{g}^{rt+r-1}\rangle + \frac{Q}{2}\eta^2 \big\|\bar{g}^{rt} + \ldots + \bar{g}^{rt+r-1}\big\|_2^2\Big]$$

$$= E\Big[\Phi\left(\bar{\theta}^{rt}\right) - \eta\left\langle g^{rt}, r\bar{g}^{rt}\right\rangle - \eta\Big\langle g^{rt}, \sum_{v=1}^{r-1} \bar{g}^{rt+v} - \bar{g}^{rt}\Big\rangle + \frac{Q}{2}\eta^2 \big\|\bar{g}^{rt} + \ldots + \bar{g}^{rt+r-1}\big\|_2^2\Big]$$

$$\leq E\Big[\Phi\left(\bar{\theta}^{rt}\right) - \eta r\|g^{rt}\|_2^2 + \eta\|g^{rt}\|_2 \big\|\sum_{v=1}^{r-1} \bar{g}^{rt+v} - \bar{g}^{rt}\big\|_2 + \frac{Q}{2}\eta^2 \big\|\bar{g}^{rt} + \ldots + \bar{g}^{rt+r-1}\big\|_2^2\Big]. \quad (17)$$

Now we bound each term in (17) using Lemmas 2 and 3. We first use the upper and lower bounds in Lemma 3 to bound the gradient norm. First, to bound the second term on the right hand side (rhs) of (17) we use the PL-inequality in (14) of Lemma 3

$$\|g^{rt}\|_2 \geq \frac{\gamma^{L-1}}{N}\underline{\Omega}_{3\to L}\underline{\Omega}_F \left\|f_L(\bar{\theta}^{rt}) - y\right\|_2. \quad (18)$$

We bound gradient norm in the third term using the upper bound of gradient in (15) of Lemma 3

$$\|g^{rt}\|_2 \leq \frac{L}{N}\frac{\bar{\Omega}_{1\to L}}{\min\limits_{l\in[L]}\bar{\Omega}_l} \left\|f_L(\bar{\theta}^{rt}) - y\right\|_2 := T_1 \quad (19)$$

Additionally, we use (11) in Lemma 2 to bound the gradient drift in the third term, we get

$$\|\sum_{v=1}^{r-1} \bar{g}^{rt+v} - \bar{g}^{rt}\|_2 \leq \eta\frac{\rho L}{N}\frac{\bar{\Lambda}_{1\to L}}{\min\limits_{l\in[L]}\bar{\Lambda}_l}\frac{A^{\frac{v+1}{2}}-1}{\sqrt{A}-1}\sqrt{\sum_{k=1}^{K} Q_k^2\|X_k\|_F^2}\|f_L(\bar{\theta}^{rt}) - y\|_2 := T_2 \quad (20)$$

Next, using (10) in Lemma 2 to bound fourth term on the rhs, the sum of stochastic gradient as

$$\big\|\bar{g}^{rt} + \ldots + \bar{g}^{rt+r-1}\big\|_2 \leq \frac{\rho L\|X\|_F}{N}\frac{A^{\frac{v+1}{2}}-1}{\sqrt{A}-1}\frac{\bar{\Lambda}_{1\to L}}{\min\limits_{l\in[L]}\bar{\Lambda}_l}\|f(\bar{\theta}^{rt}) - y\|_2 := T_3.$$

Finally, plugging the bounds for each term in (17), using the definition of loss function $\Phi(\bar{\theta}^{rt}) = \frac{1}{2N}\|f_L(\bar{\theta}^{rt}) - y\|_2^2$ along with the choice of step-size $\eta < \frac{\frac{\gamma^{2L-2}}{N^2}\underline{\Omega}_{3\to L}^2\underline{\Omega}_F^2}{2T_1T_2+QT_3^2}$ , we get

$$E[\Phi(\bar{\theta}^{r(t+1)})] \leq \left(1 - \frac{r\eta}{N}\gamma^{2(L-1)}\underline{\Omega}_{3\to L}^2\underline{\Omega}_F^2\right) E[\Phi(\bar{\theta}^{rt})]. \quad (21)$$

Using the above inequality recursively, we get the statement of Proposition 1. □

Now Step 3 is complete and we move on to define the initialization strategy of Step 4. It is important to note that Proposition 1 utilized Lemmas $1-3$, all of which impose some conditions on the singular values of the model parameters and the outputs of the first layer at each client during the entire training phase. Next, we define the initialization strategy that ensures that the conditions of Lemmas $1-3$ are satisfied almost surely.

Next, we go to Step 4, and discuss the initialization strategy. Define $\underline{\lambda}_l := \sigma_{\min}\left(W_l^0\right)$ and

$$\bar{\lambda}_l := \begin{cases} \frac{2}{3}\left(1 + \sigma_{\max}(W_l^0)\right), & \text{for } l \in \{1, 2\}, \\ \sigma_{\max}(W_l^0), & \text{for } l \in \{3, \ldots, L\} \end{cases}. \quad (22)$$

We also define the largest and smallest singular values of the output of the first layer at initialization for each client as $\underline{\alpha}_{0,k} := \sigma_{\min}\left(\sigma\left(X_k W_{1,k}^0\right)\right)$. Similarly, for the centralized setting when all the clients share the same parameter and full data, we define $\underline{\alpha}_0 := \sigma_{\min}\left(\sigma\left(X W_1^0\right)\right)$.

**Initialization Strategy:** Given any $\epsilon < \Phi(\theta^0)$, we initialize the model weights such that for some constants $M_1, M_2, M_3 > 0$, the following are satisfied

$$\frac{M_1 \min\limits_{l\in[L]}\bar{\lambda}_l}{\|X\|_F\bar{\lambda}_{1\to L}} \cdot \frac{\Phi(\theta^0)^{\frac{3}{2}}}{\epsilon} \leq \begin{cases} \frac{1}{2}\underline{\lambda}, \ l \in \{3, \ldots, L\}, \\ 1, \ l \in \{1, 2\}, \end{cases}, \quad (23)$$

$$\frac{M_2 \min\limits_{l\in[L]}\bar{\lambda}_l}{\bar{\lambda}_{1\to L}} \cdot \frac{\Phi(\theta^0)^{\frac{3}{2}}}{\epsilon} \leq \min\left(\underline{\alpha}_0, \min\limits_{k\in[K]}\underline{\alpha}_{0,k}\right), \qquad M_3\underline{\lambda}_{3\to L}\underline{\alpha}_0 \geq \frac{\bar{\lambda}_{1\to L}}{\min\limits_{l\in[L]}\bar{\lambda}_l}. \quad (24)$$

To satisfy the required initialization, we follow the initialization strategy of Nguyen and Mondelli (2020). First, randomly initialize $\left[W_1^0\right]_{ij} \sim \mathcal{N}(0, 1/d_{\text{in}}^2)$. Broadcast $\left[W_1^0\right]_{ij}$ to each client and collect $F_{1,k}$, which is the output of the first layer of each client, as well as the norm of local data $\|X_k\|_F$. With $F_{1,k}$, $\underline{\alpha}_0$ and $\underline{\alpha}_{0,k}$ can be computed. For (23), since we have $n_1 > N$, $\underline{\alpha}_0$ and $\underline{\alpha}_{0,k}$ are strictly positive. Then it is easy to verify that given $\epsilon > 0$, (23) and the second relation in (24) will be satisfied if we choose large enough $\frac{\bar{\lambda}_{1 \to L}}{\min\limits_{l \in [L]} \underline{\lambda}_l}$. This can be realized by choosing arbitrarily large $\bar{\lambda}_l$, $l \in \{3, \cdots, L\}$. In order to satisfy the first relation in (24), we need to make $\bar{\lambda}_l$ and $\underline{\lambda}_l$ close to each other. Intuitively, one way is to construct $\left(W_l^0\right)_{l=3}^L$ such that $\underline{\lambda}_l = \bar{\lambda}_l = \zeta > 1$, where $\zeta$ can be chosen to be any large number such that (23) and the second relation in (24) are satisfied. We also need to upper bound $\Phi(\theta^0)$. This can be done by choosing small $W_2^0$. Randomly initialize $W_2^0$ such that $\left[W_2^0\right]_{ij} \sim \mathcal{N}(0, \kappa)$. We can set $\kappa$ to be arbitrarily small, then $\Phi(\theta^0)$ is bounded by $\frac{2}{N}\|y\|_2^2$ with high probability (see (10) in Nguyen and Mondelli (2020)). Note that the desired error $\epsilon$ is another key constant in the initialization. When we expect the error to be small, (23) and the second relation in (24) will be more strict. But this is not an issue since we can choose a larger $\zeta$ such that the initial conditions are satisfied. The detailed initialization strategy that ensures that the conditions of Lemmas 1, 2 and 3 are satisfied is given in the Appendix B.2.

Next, let us state our main result, which indicates the linear convergence of local SGD to any $\epsilon$-optimal solution (see Definition 1). The proof is attached in Appendix B.3.

**Theorem 1.** *Using FedAvg-SGD to minimize* (3) *with Algorithm 1. Suppose Assumptions 1, 2 and 3 are satisfied, then there exists an initialization strategy such that for any $\epsilon < \Phi(\theta^0)$, there exists step-size $\eta > 0$ such that we have (where $\mu' := \frac{r}{2N} \gamma^{2(L-2)} \left(\frac{1}{2}\right)^{2(L-1)} \underline{\lambda}_{3 \to L}^2 \underline{\alpha}_0^2$, and $\eta\mu' < 1$)*

$$E[\Phi(\bar{\theta}^{r(t+1)})] \le (1 - \mu'\eta)^t \Phi(\theta^0), \; t \in \{0, \ldots, T-1\}.$$

Theorem 1 shows that, for any $\epsilon > 0$, we can always find an initialization, such that FedAvg-SG achieves an $\epsilon$ accuracy within $O\left(\log(\frac{1}{\epsilon})\right)$ rounds of communication. Notice that there is no heterogeneity assumption on the data (see Remark 2), and no assumption on the Lipschitz gradient of the loss function.

**Remark 5.** *We comment on the key novelties of this work compared to Nguyen and Mondelli (2020). (1) Our work requires a careful analysis to deal with multiple local updates at each client. Note that in contrast to Nguyen and Mondelli (2020), for our algorithm there is no guarantee that the overall objective will always decrease during local updates. In fact, our analysis demonstrates that the overall objective can increase after each local iteration, we show that this increase will be compensated by the descent in the objective value between each communication round.*

*(2) Our algorithm and analysis can deal with the stochastic gradients for conducting local updates, while Nguyen and Mondelli (2020) only considered gradient descent in a centralized setting. A key step in our analysis is to characterize the relationship between the stochastic and full gradient updates, which is illustrated in Lemma 1.*

**Remark 6.** *We comment on the choice of parameters and the convergence rate. As will be shown in Appendix B.3, by utilizing our initialization strategy, we can choose $\eta = c/\mu'$ for some constant $c \in (0, 1)$ (independent of $\epsilon$). This implies that $\mu'\eta = c < 1$, which further implies that we have $(1 - \mu'\eta) < 1$ in Theorem 1, ensuring linear convergence of FedAvg-SGD.*

Finally, we present the convergence guarantees for the case when FedAvg-GD is utilized.

**Corollary 1.** *Using FedAvg-GD to minimize* (3) *with Algorithm 1. Suppose Assumptions 2 and 3 are satisfied, then there exists an initialization strategy and step-size $\eta > 0$, such that we have*

$$\Phi(\bar{\theta}^{r(t+1)}) \le (1 - \mu'\eta)^t \Phi(\theta^0), \; \forall t \in \{0, \ldots, T-1\}. \tag{25}$$

**Remark 7.** *Corollary 1 implies that FedAvg-GD achieves linear convergence when optimizing* (3). *We note that the result of Corollary 1 is much stronger compared to Theorem 1 as the initialization for FedAvg-GD is independent of $\epsilon$ compared to the one for FedAvg-SGD (shown in Appendix B.2).*

## 5 NUMERICAL EXPERIMENTS

In this section, we analyze the effect of increasing the network sizes on popular image classification tasks with MNIST, Fashion MNIST and CIFAR-10 data sets. We compare the performance of

FedAvg in both homogeneous (i.i.d.) and heterogeneous (non-i.i.d.) data settings. Through our experiments we establish that larger sized networks uniformly outperform smaller networks under different settings. Next, we discuss the data and the model setting for our experiments.

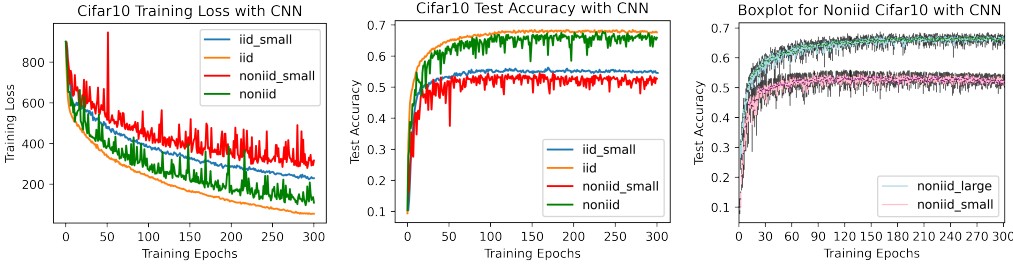

Figure 1: CIFAR-10 with CNN: FedAvg-SGD on large and small size CNN.

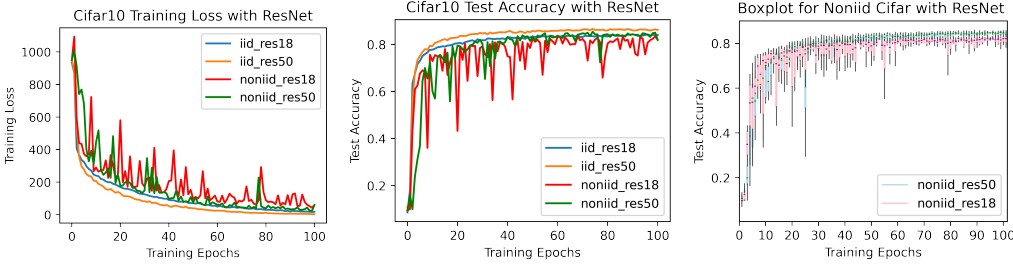

Figure 2: CIFAR-10 with ResNet: Comparison of FedAvg-SGD on ResNet18 and ResNet50.

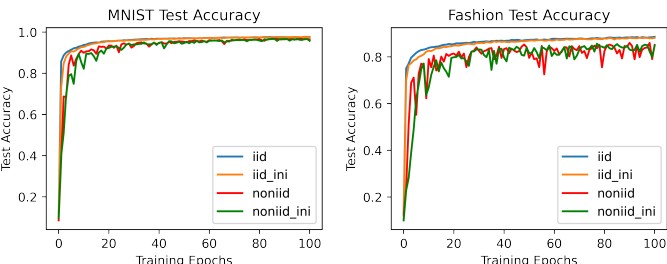

Figure 3: Test accuracy for MNIST (left) and Fashion MNIST (right) datasets. We compare the performance with (standard) random initialization and the proposed initialization strategy for both iid and noniid settings. Legends 'iid_ini' and 'noniid_ini' represent the proposed initialization strategy.

**Data set:** For MNIST, Fashion MNIST and CIFAR-10 data sets, we split the data set among $K = 100$ clients. For the homogeneous (i.i.d.) setting, we randomly distribute the complete data set with $60,000$ samples to each client. To model the heterogeneous (non i.i.d.) setting, we split the clients into two sets. One set of clients receive randomly drawn samples while the second set of clients receive data from only two out of ten labels McMahan et al. (2017). For our experiments on MNIST and Fashion MNIST data, $70\%$ of the users receive non-i.i.d samples, while for CIFAR-10 data, the fraction is $20\%$.

**Results and Discussion** For each setting, we compare the training loss and testing accuracy of FedAvg on smaller and larger sized networks. To analyze the effect of network sizes on the stability of FedAvg, we also plot the performance of FedAvg averaged over 10 iterations for non-i.i.d. client data setting for all the network architectures. From our experiments, we make a few observations. First, we observe from Figures 1 and 2 that in all the cases, the i.i.d setting has more stable performance (lower variance) than non-i.i.d setting. Second, we note that the larger network uniformly outperforms the smaller network under all the settings. Third, we note from the box plots in Figures 1 and 2 that the performance of the larger networks have lower variance, hence more stable performance compared with what can be achieved by the smaller networks. Finally, we compare the random initialization with special initialization strategy which satisfies (23), (24). We can conclude from Figure 3 that these two initialization are similar in test performance.

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

APPENDIX

## A    RELATED WORK

**Overparameterized Neural Networks.**    The surprising performance of overparameterized neural networks has raised significant research interest in the ML community to analyze the phenomenon of overparameterization Belkin et al. (2019). Consequently, a number of works have analyzed the performance of centralized (stochastic) gradient descent (S)GD on overparameterized neural network architectures under different settings Jacot et al. (2018); Li and Liang (2018); Arora et al. (2019); Du et al. (2018; 2019); Allen-Zhu et al. (2019); Zou and Gu (2019); Nguyen and Mondelli (2020); Nguyen (2021). The authors in Jacot et al. (2018), showed that an infinite width neural network when trained using gradient descent (GD) behaves like a kernel method with the kernel defined as neural tangent kernel (NTK). Using this NTK parameterization Li and Liang (2018) showed that deep neural networks trained using GD require $\Omega(N^4)$ width to find the global optimal. This result was later improved to $\Omega(N^3)$ in Huang and Yau (2020). The authors in Du et al. (2018) and Du et al. (2019) also analyze the performance of GD on overparameterized neural networks under different settings. Under standard parameterization, the work Allen-Zhu et al. (2019) studied the convergence of SGD and showed that network width of $\Omega(N^{24})$ suffices to guarantee linear convergence. Recently, Nguyen and Mondelli (2020) and Nguyen (2021) have improved the dependence on the width and have shown that GD requires only $\Omega(N)$ width to achieve linear convergence. All the works mentioned above focus on the centralized setting, and therefore, do not deal with data heterogeneity problem.

## B    PROOF OF MAIN RESULT

### B.1    PROOF OF LEMMAS

We define some additional notations before we state some lemmas which are needed in the proof. Let $\otimes$ denote the Kronecker product, and denote $\Sigma_l := \text{diag}\left[\text{vec}\left(\sigma'\left(F_{l-1}W_l\right)\right)\right] \in \mathbb{R}^{Nn_l \times Nn_l}$, $\Sigma_{l,k} := \text{diag}\left[\text{vec}\left(\sigma'\left(F_{l-1,k}W_{l,k}\right)\right)\right] \in \mathbb{R}^{N_k n_l \times N_k n_l}$ and $\tilde{\Sigma}_{l,k} := \text{diag}\left[\text{vec}\left(\sigma'\left(\tilde{F}_{l-1,k}W_{l,k}\right)\right)\right] \in \mathbb{R}^{mn_l \times mn_l}$. Define $f_{L,k}^{rt+v} := f_{L,k}(\theta_k^{rt+v}), F_{L,k}^{rt+v} := F_{L,k}(\theta_k^{rt+v}); f_L^{rt} := f_L(\bar{\theta}^{rt}), F_L^{rt} := F_L(\bar{\theta}^{rt+v}), f_L(\theta^{rt+v}) := \text{vec}(F_L^{rt+v})$.

**Lemma 4.** *Nguyen and Mondelli (2020) Suppose Assumptions 2 and Assumption 3 are satisfied. Then for $l \in [L]$ the following holds:*

1. $g_{l,k} = \dfrac{1}{N_k}\left(I_{n_l} \otimes F_{l-1,k}^T\right) \prod_{p=l+1}^{L} \Sigma_{p-1,k}\left(W_{p,k} \otimes I_{N_k}\right)(f_{L,k} - y_k),$

$$\tag{26}$$

2. $\dfrac{\partial f_{L,k}}{\partial \text{vec}(W_{l,k})} = \prod_{p=0}^{L-l-1}\left(W_{L-p,k}^T \otimes I_{N_k}\right)\Sigma_{L-t-1}\left(I_{n_{l,k}} \otimes F_{l-1,k}\right),$ $\tag{27}$

3. $\|g_{2,k}\|_2 \geq \dfrac{1}{N_k}\sigma_{\min}\left(F_{1,k}\right)\prod_{p=3}^{L}\sigma_{\min}\left(\Sigma_{p-1,k}\right)\sigma_{\min}\left(W_{p,k}\right)\|f_{L,k} - y_k\|_2,$ $\tag{28}$

4. $\|F_{l,k}\|_F \leq \|X_k\|_F \prod_{p=1}^{l}\sigma_{\max}(W_{p,k}),$ $\tag{29}$

5. $\left\|\nabla_{W_{l,k}}\Phi_k\right\|_F \leq \dfrac{1}{N_k}\|X_k\|_F \prod_{\substack{p=1 \\ p \neq l}}^{L}\sigma_{\max}(W_{p,k})\|f_{L,k} - y_k\|_2,$ $\tag{30}$

6. $\|g_k\|_2 \leq \dfrac{L\|X_k\|_F}{N}\dfrac{\prod_{l=1}^{L}\sigma_{\max}(W_{l,k})}{\min\limits_{l \in [L]}\sigma_{\max}(W_{l,k})}\prod_{l=2}^{L}\sigma_{\max}\left(\Sigma_{l-1,k}\right)\|f_{L,k} - y_k\|_2.$ $\tag{31}$

*Furthermore, given with $\theta_k^a$ and $\theta_k^b$, if $\bar{\Lambda}_l \geq \max\left(\sigma_{\max}\left(W_{l,k}^a\right), \sigma_{\max}\left(W_{l,k}^b\right)\right)$ for some scalars*
$\bar{\Lambda}_l$. *Let* $R = \prod_{p=1}^{L} \max\left(1, \bar{\Lambda}_p\right)$. *Then, for* $l \in [L]$,

7. $\left\|F_{L,k}^a - F_{L,k}^b\right\|_F \leq \sqrt{L}\|X_k\|_F \dfrac{\prod_{l=1}^{L} \bar{\Lambda}_l}{\min_{l \in [L]} \bar{\Lambda}_l} \left\|\theta_k^a - \theta_k^b\right\|_2$,

$$(32)$$

8. $\left\|\dfrac{\partial f_L\left(\theta_k^a\right)}{\partial \operatorname{vec}\left(W_l^a\right)} - \dfrac{\partial f_L\left(\theta_k^b\right)}{\partial \operatorname{vec}\left(W_l^b\right)}\right\|_2 \leq \sqrt{L}\|X_k\|_F R\left(1 + L\beta\|X_k\|_F R\right)\left\|\theta_k^a - \theta_k^b\right\|_2 . e$ $\quad(33)$

The above Lemma follows Lemma 4.1 Nguyen and Mondelli (2020): (26) gives the expression of the vectorized gradient; (27) provides the vectorized Jacobian matrix of the output of the network; (28) gives a lower bound on the norm of the gradient, which holds under Assumption 2. (29) provides an upper bound on the norm of output of each layer while (30) gives an upper bound on the norm of gradient of each layer; (32) derives the Lipschitz constant of the networks and (33) provides the Lipschitz constant for the Jacobian of each layer. Similar results can be derived in centralized optimization problem, so we do not include the results here.

**Lemma 5.** *(Nguyen and Mondelli, 2020, Lemma 4.3) Let* $f : \mathbb{R}^n \to \mathbb{R}$ *be a* $C^2$ *function. Let* $x, y \in \mathbb{R}^n$ *be given, and assume that* $\|\nabla f(z) - \nabla f(x)\|_2 \leq C\|z - x\|_2$ *for every* $z = x + t(y - x)$ *with* $t \in [0, 1]$. *Then,*

$$f(y) \leq f(x) + \langle \nabla f(x), y - x \rangle + \frac{C}{2}\|x - y\|^2.$$

**Lemma 6.** *For constant* $C, \mu, \rho$, *if* $\eta \to 0$, *we have*

$$\lim_{\eta \to 0} \sqrt{1 + 3\rho C \eta}^{\frac{1}{\log \frac{1}{1 - \mu C \eta}}} = e^{\frac{3\rho}{2\mu}} \tag{34}$$

*Furthermore, given* $\epsilon < \Phi(\theta^0)$, *let* $T = \left\lfloor \dfrac{\log\left(\Phi(\theta^0)/\epsilon\right)}{\log\left(\frac{1}{1 - \mu C \eta}\right)} + 1 \right\rfloor$, *then there exists constant* $\xi$, *such that*

$$\sup_{0 < \eta < \min\left(\frac{1}{\rho C}, \frac{1}{\mu C}\right)} \left(\sqrt{1 + 3\rho C \eta}\right)^T \leq \frac{\xi \Phi(\theta^0)}{\epsilon}, \tag{35}$$

*where* $\xi \geq e^{\frac{3\rho}{2\mu}}$ *is a constant dependent on* $\rho$ *and* $\mu$.

*Proof.* Take logarithm on both sides, we get

$$\log\left(\sqrt{1 + 3\rho C \eta}^{\frac{1}{\log \frac{1}{1 - \mu C \eta}}}\right) = -\frac{\log(\sqrt{1 + 3\rho C \eta})}{\log(1 - \mu C \eta)} = -\frac{1}{2} \cdot \frac{\log(1 + 3\rho C \eta)}{\log(1 - \mu C \eta)} \tag{36}$$

Now let $\eta \to 0$, by L'Hôpital's rule, take derivative over $\eta$, we have

$$\lim_{\eta \to 0} -\frac{1}{2} \cdot \frac{\log(1 + 3\rho C \eta)}{\log(1 - \mu C \eta)} = \lim_{\eta \to 0} \frac{1}{2} \cdot \frac{3\rho C}{\mu C} \frac{1 - \mu C \eta}{1 + 3\rho C \eta} = \frac{3\rho}{2\mu}. \tag{37}$$

Next, if we can show the function of $\eta$, which is $\sqrt{1 + 3\rho C \eta}^{\frac{1}{\log \frac{1}{1 - \mu C \eta}}}$, has a limit when $\eta \to \min(\frac{1}{\rho C}, \frac{1}{\mu C})$, then by the continuity, it has an upper bound in $\left(0, \min(\frac{1}{\rho C}, \frac{1}{\mu C})\right)$, denote it as $\xi$. It is easy to derive that

$$\lim_{\eta \to \min\left(\frac{1}{\rho C}, \frac{1}{\mu C}\right)} \sqrt{1 + 3\rho C \eta}^{\frac{1}{\log \frac{1}{1 - \mu C \eta}}} = \lim_{\eta \to \min\left(\frac{1}{\rho}, \frac{1}{\mu}\right)} \sqrt{1 + 3\rho \eta}^{\frac{1}{\log \frac{1}{1 - \mu \eta}}} \begin{cases} 2^{\frac{1}{\log \frac{1}{1 - \frac{\mu}{\rho}}}}, & \rho > \mu, \\ 1, & \rho \leq \mu. \end{cases}$$

Then by the continuity of the function, $\left(\sqrt{1+3\rho C\eta}\right)^{\frac{1}{\log\frac{1}{1-\mu C\eta}}}$ is bounded by some constant $\xi$. Then we can derive

$$\sup_{\eta\in\left(0,\min(\frac{1}{\rho C},\frac{1}{\mu C})\right)} \sqrt{1+3\rho C\eta}^T \geq \lim_{\eta\to 0} \sqrt{1+3\rho\eta}^{\frac{\log\left(\Phi(\theta^0)/\epsilon\right)}{\log\frac{1}{1-\mu\eta}}} = e^{\frac{3\rho}{2\mu}} \cdot \frac{\Phi(\theta^0)}{\epsilon} \tag{38}$$

then we have there exists some constant $\xi \geq e^{\frac{3\rho}{2\mu}}$, such that

$$\sup_{\eta\in\left(0,\min(\frac{1}{\rho C},\frac{1}{\mu C})\right)} \sqrt{1+3\rho C\eta}^T \leq \frac{\xi\Phi\left(\theta^0\right)}{\epsilon}. \tag{39}$$

$\square$

**Lemma 7.** *Let Assumption 2 and Assumption 3 hold. For $\theta_k$, suppose there exists constant $\bar{\Lambda}_l$, $\underline{\Lambda}_l$, $\underline{\Lambda}_F$ such that and*

$$\begin{cases} \sigma_{\max}(W_{l,k}) \leq \bar{\Lambda}_l, \ l \in [L], \ k \in [K], \\ \sigma_{\min}(W_{l,k}) \geq \underline{\Lambda}_l, \ l \in \{3,\ldots,L\}, \ k \in [K], \\ \sigma_{\min}(F_{1,k}) \geq \underline{\Lambda}_F, \ k \in [K]. \end{cases} \tag{40}$$

*then we have*

$$\rho(\theta_k) \leq \frac{LN\frac{\bar{\Lambda}_{1\to L}}{\min\limits_{l\in[L]}\bar{\Lambda}_l}}{m\gamma^{L-2}\underline{\Lambda}_{3\to L}\underline{\Lambda}_F} \tag{41}$$

*Proof.* By definition, we have

$$\rho(\theta_k) = \frac{\|\tilde{g}_k\|_2}{\|g_k\|_2} \leq \frac{\|\tilde{g}_k\|_2}{\|g_{2,k}\|_2}. \tag{42}$$

Since by (31) and (28) in Lemma 4, we have

$$\|\tilde{g}_k\|_2 \leq \frac{L\|\tilde{X}_k\|_F}{m} \frac{\bar{\Lambda}_{1\to L}}{\min\limits_{l\in[L]}\bar{\Lambda}_l}\|\tilde{f}_{L,k}(\theta) - \tilde{y}\|_2, \tag{43}$$

$$\|g_{2,k}\|_2 \geq \frac{1}{N_k}\gamma^{L-2}\underline{\Lambda}_{3\to L}\underline{\Lambda}_F\|f_{L,k}(\theta) - y\|, \tag{44}$$

where $X_k$ is the sampled data at $\theta_k$. So we can derive

$$\rho(\theta_k) \leq \frac{\frac{L\|\tilde{X}_k\|_F}{m}\frac{\bar{\Lambda}_{1\to L}}{\min\limits_{l\in[L]}\bar{\Lambda}_l}\|\tilde{f}_{L,k}(\theta) - \tilde{y}_k\|_2}{\frac{1}{N_k}\gamma^{L-2}\underline{\Lambda}_{3\to L}\underline{\Lambda}_F\|f_{L,k}(\theta) - y_k\|_2} \leq \frac{LN\|X\|_F\frac{\bar{\Lambda}_{1\to L}}{\min\limits_{l\in[L]}\bar{\Lambda}_l}}{m\gamma^{L-2}\underline{\Lambda}_{3\to L}\underline{\Lambda}_F}, \tag{45}$$

where the last inequality is because $\|\tilde{X}_k\|_F \leq \|X\|_F$ and $\|\tilde{f}_{L,k}(\theta) - \tilde{y}_k\|_2 \leq \|f_{L,k}(\theta) - y_k\|_2$. $\square$

**Lemma 8.** *For the FedAvg-SGD algorithm, given step size $\eta > 0$, $v \in \{0,1,\ldots,r-1\}$ and $q \in \{0,1,\ldots,v-1\}$. Suppose the following conditions hold:*

1.$\bar{\Lambda}_l \geq \sup_{k\in[K]} \sigma_{\max}\left(W_{l,k}^{rt+q}\right),$

$$\tag{46}$$

2.$\rho \geq \sup_{k\in[K]} \rho\left(\theta_k^{rt+q}\right),$ $\tag{47}$

3.$\Phi_k(\theta^{rt+q}) \leq A^q \cdot \Phi_k(\theta^{rt}), \ k \in [K],$ $\tag{48}$

*then we have*

$$\left\|\sum_{q=0}^{v} \bar{\tilde{g}}^{rt+q}\right\|_2 \leq \sum_{q=0}^{v} \left\|\bar{\tilde{g}}^{rt+q}\right\|_2 \leq \frac{\rho L\|X\|_F}{N}\frac{A^{\frac{v+1}{2}}-1}{\sqrt{A}-1}\frac{\bar{\Lambda}_{1\to L}}{\min\limits_{l\in[L]}\bar{\Lambda}_l}\|f_L^{rt} - y\|_2. \tag{49}$$

*Further, there exists constant $Q_k$, such that $\forall k \in [K]$ we have*

$$\left\| g_k^{rt+q+1} - g_k^{rt+q} \right\|_2 \le \rho Q_k \left\| \theta_k^{rt+q+1} - \theta_k^{rt+q} \right\|_2,$$

(50)

*and*

$$\left\| \bar{g}^{rt+v} - \bar{g}^{rt} \right\|_2 \le \sum_{q=0}^{v-1} \left\| \bar{g}^{rt+q+1} - \bar{g}^{rt+q} \right\|_2 \le \frac{\eta \rho L}{N} \frac{\Lambda_{1 \to L}}{\min_{l \in [L]} \Lambda_l} \frac{A^{\frac{v+1}{2}} - 1}{\sqrt{A} - 1} \sqrt{\sum_{k=1}^{K} Q_k^2 \|X_k\|_F^2} \|f_L^{rt} - y\|_2.$$

(51)

*Proof.* First, let us show (49).

$$\| \sum_{q=0}^{v} \bar{\tilde{g}}^{rt+v} \|_2 \overset{(i)}{\le} \sum_{q=0}^{v} \|\bar{\tilde{g}}^{rt+v}\|_2 \overset{(ii)}{\le} \sum_{q=0}^{v} \sum_{k=1}^{K} \frac{N_k}{N} \|\tilde{g}_k^{rt+v}\|_2 \overset{(iii)}{\le} \rho \sum_{q=0}^{v} \sum_{k=1}^{K} \frac{N_k}{N} \|g_k^{rt+v}\|_2$$

(52)

$$\overset{(iv)}{\le} \frac{\rho L}{N} \sum_{q=0}^{v} \sum_{k=1}^{K} \|X_k\|_F \frac{\bar{\Lambda}_{1 \to L}}{\min_{l \in [L]} \bar{\Lambda}_l} \|f_{L,k}^{rt+q} - y\|_2$$

(53)

$$= \frac{\rho L}{N} \frac{\bar{\Lambda}_{1 \to L}}{\min_{l \in [L]} \bar{\Lambda}_l} \sum_{k=1}^{K} \|X_k\|_F \sum_{q=0}^{v} \|f_{L,k}^{rt+q} - y_k\|_2$$

(54)

$$\overset{(v)}{\le} \frac{\rho L}{N} \frac{\bar{\Lambda}_{1 \to L}}{\min_{l \in [L]} \bar{\Lambda}_l} \sum_{k=1}^{K} \|X_k\|_F \sum_{q=0}^{v} A^{\frac{q}{2}} \|f_{L,k}^{rt+q} - y_k\|_2$$

(55)

$$= \frac{\rho L}{N} \frac{\bar{\Lambda}_{1 \to L}}{\min_{l \in [L]} \bar{\Lambda}_l} \frac{A^{\frac{v+1}{2}} - 1}{\sqrt{A} - 1} \sum_{k=1}^{K} \|X_k\|_F \|f_{L,k}^{rt+q} - y_k\|_2$$

(56)

$$\overset{(vi)}{\le} \frac{\rho L}{N} \frac{\bar{\Lambda}_{1 \to L}}{\min_{l \in [L]} \bar{\Lambda}_l} \frac{A^{\frac{v+1}{2}} - 1}{\sqrt{A} - 1} \sqrt{\sum_{k=1}^{K} \|X_k\|_F^2} \sqrt{\sum_{k=1}^{K} \|f_{L,k}^{rt+q} - y_k\|_F^2}$$

(57)

$$= \frac{\rho L \|X\|_F}{N} \frac{\bar{\Lambda}_{1 \to L}}{\min_{l \in [L]} \bar{\Lambda}_l} \frac{A^{\frac{v+1}{2}} - 1}{\sqrt{A} - 1} \|f_L^{rt} - y\|_2,$$

(58)

So we can derive (49). Next, we show (50). Let us denote $Jf_{L,k}^{rt+q} := \left[ \frac{\partial f_{L,k}^{rt+q}}{\partial \operatorname{vec}(W_{1,k})}, \dots, \frac{\partial f_{L,k}^{rt+q}}{\partial \operatorname{vec}(W_{L,k})} \right]$. By triangle inequality, we have

$$\left\| g_k^{rt+q+1} - g_k^{rt+q} \right\|_2 = \left\| Jf_{L,k}^{rt+q+1} \left( f_{L,k}^{rt+q+1} - y_k \right) - Jf_{L,k}^{rt+q} \left( f_{L,k}^{rt+q} - y_k \right) \right\|$$

$$\le \left\| f_{L,k}^{rt+q+1} - f_{L,k}^{rt+q} \right\|_2 \left\| Jf_{L,k}^{rt+q+1} \right\|_2 + \left\| Jf_{L,k}^{rt+q+1} - Jf_{L,k}^{rt+q} \right\|_2 \left\| f_{L,k}^{rt+q} - y_k \right\|_2$$

(59)

Now we find the bound for each term in (59). Since $\max \left( \sigma_{\max} \left( W_{l,k}^{rt+q+1} \right), \sigma_{\max} \left( W_{l,k}^{rt+q} \right) \right) \le \bar{\Lambda}_l$, by (32) in Lemma 4, we get

$$\left\| f_{L,k}^{rt+q+1} - f_{L,k}^{rt+q} \right\|_2 \le \sqrt{L} \|X_k\|_F \frac{\bar{\Lambda}_{1 \to L}}{\min_{l \in [L]} \bar{\Lambda}_l} \left\| \theta_k^{rt+q+1} - \theta_k^{rt+q} \right\|_2$$

(60)

Further, by (27) we have

$$\left\| Jf_{L,k}^{rt+q+1} \right\|_2 \le \sum_{l=1}^{L} \left\| \frac{\partial Jf_{L,k}^{rt+q+1}}{\partial \operatorname{vec}(W_{l,k})} \right\|_2 \le L \|X_k\|_F \frac{\bar{\Lambda}_{1 \to L}}{\min_{l \in [L]} \bar{\Lambda}_l}.$$

(61)

Using (33) in Lemma 4, we have

$$
\begin{aligned}
\|Jf_{L,k}^{rt+q+1} - Jf_{L,k}^{rt+q}\|_2 &\leq \sum_{l=1}^{L} \left\| \frac{\partial Jf_{L,k}^{rt+q+1}}{\partial \mathrm{vec}\,(W_{l,k})} - \frac{\partial Jf_{L,k}^{rt+q}}{\partial\,\mathrm{vec}\,(W_{l,k})} \right\|_2 \\
&\leq L^{\frac{3}{2}} \|X_k\|_F R' \left(1 + L\beta \|X_k\|_F R'\right) \left\|\theta_k^{rt+q+1} - \theta_k^{rt+q}\right\|_2,
\end{aligned} \tag{62}
$$

where $R' = \prod_{p=1}^{L} \max\left(1, \bar{\Lambda}_l\right)$. So plug the above bounds into (59). Set Lipschitz constant

$$
Q_k = \frac{L\sqrt{L}}{N_k} \|X_k\|_F^2 \frac{\bar{\Lambda}_{1\to L}^2}{\min_{l\in[L]} \bar{\Lambda}_l^2} + \frac{L\sqrt{L}}{N_k} \|X_k\|_F \left(1 + L\beta \|X_k\|_F R'\right) R' \left\|f_{L,k}^0 - y_k\right\|_2, \tag{63}
$$

then we derive

$$
\left\|g_k^{rt+q+1} - g_k^{rt+q}\right\|_2 \leq Q_k \|\theta_k^{rt+q+1} - \theta_k^{rt+q}\|_2. \tag{64}
$$

Now (50) is proved. Last, we prove (51). We have

$$
\begin{aligned}
\left\|\bar{g}^{rt+v} - \bar{g}^{rt}\right\|_2 &\leq \sum_{q=0}^{v-1} \left\|\bar{g}^{rt+q+1} - \bar{g}^{rt+q}\right\|_2 \overset{(i)}{\leq} \sum_{q=0}^{v-1} \sum_{k=1}^{K} \frac{N_k}{N} \left\|g_k^{rt+q+1} - g_k^{rt+q}\right\|_2 \\
&\overset{(ii)}{\leq} \sum_{q=0}^{v-1} \sum_{k=1}^{K} \frac{N_k}{N} Q_k \left\|\theta_k^{rt+q+1} - \theta_k^{rt+q}\right\|_2 = \sum_{q=0}^{v-1} \sum_{k=1}^{K} \frac{N_k}{N} Q_k \cdot \eta \left\|\tilde{g}_k^{rt+q}\right\|_2 \\
&\overset{(iii)}{\leq} \sum_{q=0}^{v-1} \sum_{k=1}^{K} \frac{Q_k}{N} L \|X_k\|_F \frac{\bar{\Lambda}_{1\to L}}{\min_{l\in[L]} \bar{\Lambda}_l} \left\|f_L^{rt+q} - y_k\right\|_2 \\
&\overset{(iv)}{\leq} \frac{\eta L}{N} \frac{\bar{\Lambda}_{1\to L}}{\min_{l\in[L]} \bar{\Lambda}_l} \sum_{k=1}^{K} Q_k \|X_k\|_F \sum_{q=0}^{v-1} A^{\frac{q}{2}} \left\|f_L^{rt+q} - y_k\right\|_2^2 \\
&\leq \frac{\eta L}{N} \frac{\bar{\Lambda}_{1\to L}}{\min_{l\in[L]} \bar{\Lambda}_l} \frac{A^{\frac{v+1}{2}} - 1}{\sqrt{A} - 1} \sum_{k=1}^{K} Q_k \|X_k\|_F \left\|f_L^{rt+q} - y_k\right\|_2^2 \\
&\overset{(v)}{\leq} \frac{\eta L}{N} \frac{\bar{\Lambda}_{1\to L}}{\min_{l\in[L]} \bar{\Lambda}_l} \frac{A^{\frac{v+1}{2}} - 1}{\sqrt{A} - 1} \sqrt{\sum_{k=1}^{K} Q_k^2 \|X_k\|_F^2} \sqrt{\sum_{k=1}^{K} \left\|f_L^{rt+q} - y_k\right\|_2^2} \\
&= \frac{\eta L}{N} \frac{\bar{\Lambda}_{1\to L}}{\min_{l\in[L]} \bar{\Lambda}_l} \frac{A^{\frac{v+1}{2}} - 1}{\sqrt{A} - 1} \sqrt{\sum_{k=1}^{K} Q_k^2 \|X_k\|_F^2} \cdot \|f_L^{rt+q} - y\|_2,
\end{aligned}
$$

where (i) uses triangle inequality; (ii) uses the Lipschitz gradient assumption in condition 2; (iii) comes from (31) in Lemma 4; (iv) uses condition 3; (v) is from Cauchy-Schwartz inequality. $\qquad \square$

## B.2 INITIALIZATION STRATEGY

**Detailed Initialization for FedAvg-SGD:** Denote

$$
P := \frac{L\|X\|_F}{N}\left(\frac{7}{4}\right)^{L-1}(2^r - 1),
$$

$$
C := PL\|X\|_F\left(\frac{3}{2}\right)^{L-1}\frac{\bar{\lambda}_{1\to L}^2}{\min_{l\in[L]}\bar{\lambda}_l^2}, \tag{65}
$$

$$
\rho := \frac{LN\|X\|_F 7^{L-1}\frac{\bar{\lambda}_{1\to L}}{\min_{l\in[L]}\bar{\lambda}_l}}{m\gamma^{L-2}\underline{\lambda}_{3\to L}\min\left(\underline{\alpha}_0, \min_{k\in[K]}\underline{\alpha}_{0,k}\right)}, \tag{66}
$$

$$
\mu := \frac{\frac{r}{2N^2}\gamma^{2(L-2)}\left(\frac{1}{2}\right)^{2(L-1)}\underline{\lambda}_{3\to L}^2\underline{\alpha}_0^2}{C}. \tag{67}
$$

Suppose given any small $\epsilon$ such that $\epsilon < \Phi(\theta^0)$, the initialized weights satisfies the following conditions:

$$
\frac{2N^{\frac{3}{2}}}{L\|X\|_F(\frac{3}{2})^{L-1}\frac{\bar{\lambda}_{1\to L}}{\min_{l\in[L]}\bar{\lambda}_l}}\cdot\frac{\xi\Phi(\theta^0)}{\epsilon}\sqrt{2\Phi(\theta^0)} \leq \left\{\begin{array}{l}\frac{1}{2}\underline{\lambda}_l,\ l\in\{3,\cdots,L\},\\ 1,\ l\in\{1,2\}.\end{array}\right. \tag{68}
$$

$$
\frac{2N^{\frac{3}{2}}}{L(\frac{3}{2})^{L-1}\frac{\bar{\lambda}_{1\to L}}{\min_{l\in[L]}\bar{\lambda}_l}}\cdot\frac{\xi\Phi(\theta^0)}{\epsilon}\sqrt{2\Phi(\theta^0)} \leq \frac{1}{2}\min\left(\underline{\alpha}_0, \min_{k\in[K]}\underline{\alpha}_{0,k}\right). \tag{69}
$$

where $\xi \geq e^{\frac{3\rho}{2\mu}}$ is some constant dependent on $\rho$ and $\mu$.

Now we provide a detailed way to realize the above initialization condition. To satisfy the required initialization, we follow the initialization strategy of Nguyen and Mondelli (2020). First, randomly initialize $[W_1^0]_{ij} \sim \mathcal{N}(0, 1/d_{\text{in}}^2)$. Broadcast $[W_1^0]_{ij}$ to each client and collect $F_{1,k}$, which is the output of the first layer of each client, as well as the norm of local data $\|X_k\|_F$ and norm of local label $\|y_k\|_2$. With $F_{1,k}$, $\underline{\alpha}_0$ and $\underline{\alpha}_{0,k}$ can be computed. For (23), since we have $n_1 > N$, $\underline{\alpha}_0$ and $\underline{\alpha}_{0,k}$ are strictly positive with probability 1. Then it is easy to verify that given $\epsilon > 0$, (23) and the second relation in (24) will be satisfied if we choose large enough $\frac{\bar{\lambda}_{1\to L}}{\min_{l\in[L]}\bar{\lambda}_l}$. This can be realized by choosing arbitrarily large $\lambda_l$, $l \in \{3,\cdots,L\}$. However, notice that by Lemma 6, the constant $\xi$, which is defined in (39), is only dependent on $\rho$ and $\mu$ and $\xi \geq e^{\frac{3\rho}{2\mu}}$. So if we can fix $\rho$ and $\mu$ as some constants, $\xi$ is a bounded constant. Notice in (66) and (67), for $l \in \{3,\cdots,L\}$, if we can make $\bar{\lambda}_l$ and $\underline{\lambda}_l$ close to each other, then $\rho$ and $\mu$ are also close, so $\frac{3\rho}{2\mu}$ is not large. This is equivalent to the first relation in (24) in main text. In order to satisfy the above conditions, one way is to construct $\left(W_l^0\right)_{l=3}^L$ in such way that $\underline{\lambda}_l = \bar{\lambda}_l = \zeta > 1$, where $\zeta$ can be chosen to be any large number such that (23) and the second relation in (24) are satisfied. Specifically, we can utilize the following construction: Initialize $W_l^0$ such that its top block is a scaled identity matrix and rest of entries are zero

$$
W_l^0 = \left[\begin{array}{c}\zeta\cdot I_{n_l}\\ \mathbf{0}\end{array}\right] \in \mathbb{R}^{n_l\times n_{l-1}},\ l = 3,\ldots,L. \tag{70}
$$

We also need to upper bound $\Phi(\theta^0)$. This can be done by choosing small $W_2^0$. Randomly initialize $W_2^0$ such that $\left[W_2^0\right]_{ij} \sim \mathcal{N}(0, \kappa)$. We can set $\kappa$ to be arbitrarily small, similar to (10) in Nguyen

and Mondelli (2020),we can find a bound for $\Phi(\theta^0)$ with high probability:

$$\sqrt{2N\Phi(\theta^0)} = \|F_L(\theta^0) - y\|_F \tag{71}$$
$$\leq \|y\|_2 + \|F_L(\theta^0)\|_F$$
$$\leq \|y\|_2 + \prod_{l=1}^{L} \sigma_{\max}(W_l^0)\|X\|_F$$
$$\leq 2\|y\|_2 \tag{72}$$

Then the loss function at initialization can be bounded by constant $\sqrt{2N\Phi(\theta^0)} \leq 2\|y\|_2$.

**Initialization for FedAvg-GD:** The initialized weight matrices satisfy the following conditions:

$$\frac{2N\left(\left(\frac{3}{2}\right)^{L-1} + 2^{L-1}(r-1)\right)\|X\|_F}{r\gamma^{2(L-2)}\left(\frac{1}{2}\right)^{2(L-1)^2}\underline{2}_{3\to L}^2\underline{\alpha}_0^2} \cdot \frac{\bar{\lambda}_{1\to L}}{\bar{\lambda}_l} \leq \begin{cases} \frac{1}{2}\underline{\lambda}_l, l \in \{3, \cdots, L\} \\ 1, l \in \{1, 2\} \end{cases} \tag{73}$$

$$\frac{2N\left(\left(\frac{3}{2}\right)^{L-1} + 2^{L-1}(r-1)\right)\|X\|_F^2}{r\gamma^{2(L-2)}\left(\frac{1}{2}\right)^{2(L-1)^2}\underline{\lambda}_{3\to L}^2\underline{\alpha}_0^2} \cdot \bar{\lambda}_{2\to L} \leq \frac{1}{2}\underline{\alpha}_0. \tag{74}$$

The initialization strategy is similar to FedAvg-SGD, so we omit the discussion here.

## B.3 PROOF OF THEOREM 1

**Theorem 1.** *Using FedAvg-SGD to minimize* (3) *with Algorithm 1. Suppose Assumptions 1, 2 and 3 are satisfied, then there exists an initialization strategy such that for any $\epsilon < \Phi(\theta^0)$, there exists step-size $\eta > 0$ such that we have*

$$E[\Phi(\bar{\theta}^{r(t+1)})] \leq \left(1 - \mu'\eta\right)^t \Phi(\theta^0), \ t \in \{0, \ldots, T-1\} \tag{75}$$

*where $\mu' = \frac{r}{N}\gamma^{2(L-2)}\left(\frac{1}{2}\right)^{2(L-1)}\underline{\lambda}_{3\to L}^2\underline{\alpha}_0^2$.*

*Proof.* First, we provide a structure of our proof. We will show the following recursively at each communication round: 1) The averaged weights are bounded at each communication round; 2) The divergence of loss function (3) is bounded at each communication round; 3) The expected loss function (3) decreases linearly at each communication round. Further, we will show that in each local epoch within a fixed communication round, we have: 1) The weights of each client are bounded; 2) The divergence of loss function $\Phi_k$ of each client is bounded.

Now let us set $T = \left\lfloor \frac{\log\left(\Phi(\theta^0)/\epsilon\right)}{\log\left(\frac{1}{1-\mu C\eta}\right)} + 1 \right\rfloor$. If we can show (75) holds for $t = 0, \ldots, T$, then it is easy to show that

$$E[\Phi(\bar{\theta}^{rT})] \leq \left(1 - \mu C\eta\right)^T \Phi(\theta^0) \leq \epsilon.$$

We prove Theorem 1 by induction. Define

$$\rho^{rt+v} := \sup_{\substack{k\in[K] \\ q\in\{0,1,\ldots,v\}}} \rho(\theta_k^{rt+q})$$

$$\rho := \frac{Lm\|X\|_F 7^{L-1}\frac{\bar{\lambda}_{1\to L}}{\min_{l\in[L]}\underline{\lambda}_l}}{N\gamma^{L-2}\underline{\lambda}_{3\to L}\min\left(\underline{\alpha}_0, \min_{k\in[K]}\underline{\alpha}_{0,k}\right)},$$

$$\tag{76}$$

We show that $\forall t \leq T$, we have

$$
\begin{cases}
\sigma_{\max}\left(\bar{W}^{ru}\right) \leq \frac{3}{2}\bar{\lambda}_l \ u \in \{0,\dots,t\}, \ l \in [L], \\
\sigma_{\min}\left(\bar{W}^{ru}\right) \geqslant \frac{1}{2}\underline{\lambda}_l, \ u \in \{0,\dots,t\}, \ l \in \{3,\dots,L\}, \\
\sigma_{\min}\left(F_1^{ru}\right) \geqslant \frac{1}{2}\underline{\alpha}_0, \ u \in \{0,\dots,t\}, \\
\sigma_{\min}\left(F_{1,k}^{ru}\right) \geq \frac{1}{2}\underline{\alpha}_{0,k}, \ u \in \{0,\dots,t\}, \ k \in [K], \\
\rho^{rt} \leq \rho, \\
\Phi\left(\bar{\theta}^{ru}\right) \leqslant (1+3\rho C\eta)^u \, \Phi\left(\theta^0\right), \ u \in \{0,\dots,t\} \\
E\left[\Phi\left(\bar{\theta}^{ru}\right)\right] \leq (1-\mu C\eta)^u \Phi\left(\theta^0\right), \ u \in \{0,\dots,t\}
\end{cases}
\tag{77}
$$

where $\bar{\lambda}_l$ is defined in (22) and $\underline{\lambda}_l$ is the smallest eigen value of the weight matrix, $C, \mu, \rho$ defined in B.2 and $\mu C = \mu'$.

The above recursive equation describes the weight matrix and loss function in each communication round. To prove (77), we decompose the recursive equation into two steps, as follows
Step1: For a fixed $t$ and $v \in [r-1]$, given

$$
\begin{cases}
\sigma_{\max}\left(\bar{W}_l^{ru}\right) \leqslant \frac{3}{2}\bar{\lambda}_l, \ u \in \{0,\dots,t\}, \ l \in [L], \\
\sigma_{\min}\left(\bar{W}_l^{ru}\right) \geqslant \frac{1}{2}\underline{\lambda}_l, \ u \in \{0,\dots,t\}, \ l \in [L], \\
\sigma_{\min}\left(F_1^{ru}\right) \geqslant \frac{1}{2}\underline{\alpha}_0, \ u \in \{0,\dots,t\}, \\
\rho^{rt} \leq \rho; \\
\Phi\left(\bar{\theta}^{ru}\right) \leq (1+3\rho C\eta)^u \Phi\left(\theta^0\right), \ u \in \{0,\dots,t\} \\
E\left[\Phi\left(\bar{\theta}^{ru}\right)\right] \leq (1-\mu C\eta)^u \Phi\left(\theta^0\right), \ u \in \{0,\dots,t\} \\
\Phi_k\left(\theta_k^{rt+q}\right) \leq (1+3\rho C'\eta)^q \Phi_k\left(\theta_k^{rt}\right), \ q \in \{0,\dots,v-1\}, \ k \in [K], \\
\sigma_{\max}\left(W_{l,k}^{rt+q}\right) \leq \frac{7}{4}\bar{\lambda}_l, \ q \in \{0,\dots,v-1\}, \ l \in [L], \ k \in [K], \\
\sigma_{\min}\left(W_{l,k}^{rt+q}\right) \leq \frac{1}{4}\underline{\lambda}_l, \ q \in \{0,\dots,v-1\}, \ l \in [L], \ k \in [K], \\
\sigma_{\min}\left(F_{1,k}^{rt+q}\right) \geq \frac{1}{4}\underline{\alpha}_{0,k}, \ q \in \{0,\dots,v-1\}, \ k \in [K], \\
\rho^{rt+v-1} \leq \rho,
\end{cases}
\tag{78}
$$

we aim to show

$$
\begin{cases}
\sigma_{\max}\left(W_{l,k}^{rt+q}\right) \leq \frac{7}{4}\bar{\lambda}_l, \ q \in \{0,\dots,v\}, \ l \in [L], \ k \in [K], \\
\sigma_{\min}\left(W_{l,k}^{rt+q}\right) \geq \frac{1}{4}\underline{\lambda}_l, \ q \in \{0,\dots,v\}, \ l \in [L], \ k \in [K], \\
\sigma_{\min}\left(F_{1,k}^{rt+q}\right) \geq \frac{1}{4}\underline{\alpha}_{0,k}, \ q \in \{0,\dots,v\}, \ k \in [K] \\
\rho^{rt+v} \leq \rho, \\
\Phi_k\left(\theta_k^{rt+q}\right) \leq (1+3\rho C'\eta)^q \Phi_k\left(\theta_k^{rt}\right), \ q \in \{0,1,\dots,v\}, \quad k \in [K].
\end{cases}
\tag{79}
$$

where $C' = \max_k \left(\frac{1}{N_k}\left(\frac{7}{4}\right)^{2(L-1)} \frac{\bar{\lambda}_{1\to L}^2}{\min_{l\in[L]}\underline{\lambda}_l^2}\right)$.
Step 2: Given (78) and (79), we show

$$
\begin{cases}
\sigma_{\max}\left(\bar{W}_l^{ru}\right) \leqslant \frac{3}{2}\bar{\lambda}_l, \ u \in \{0,\dots,t+1\}, \ l \in [L], \\
\sigma_{\min}\left(\bar{W}_l^{ru}\right) \geqslant \frac{1}{2}\underline{\lambda}_l, \ u \in \{0,\dots,t+1\}, \ l \in [L], \\
\sigma_{\min}\left(F_1^{ru}\right) \geqslant \frac{1}{2}\underline{\alpha}_0, \ u \in \{0,\dots,t+1\}, \\
\sigma_{\min}\left(F_{1,k}\right) \geq \frac{1}{2}\underline{\alpha}_{0,k}, \ u \in \{0,\dots,t+1\}, \ k \in [K], \\
\rho^{r(t+1)} \leq \rho, \\
\Phi\left(\bar{\theta}^{ru}\right) \leq (1+3\rho C\eta)^u \Phi\left(\theta^0\right), \ u \in \{0,\dots,t+1\}, \\
E\left[\Phi\left(\bar{\theta}^{ru}\right)\right] \leq (1-\mu C\eta)^u \Phi\left(\theta^0\right), \ u \in \{0,\dots,t+1\}.
\end{cases}
\tag{80}
$$

Now we show Step 1 first.
(1) We first show

$$
\begin{cases}
\sigma_{\max}\left(W_{l,k}^{rt+q}\right) \leqslant \frac{7}{4}\bar{\lambda}_l, \ q \in \{0,\dots,v\}, \ l \in [L], \ k \in [K], \\
\sigma_{\min}\left(W_{l,k}^{rt+q}\right) \geqslant \frac{1}{4}\underline{\lambda}_l, \ q \in \{0,\dots,v\}, \ l \in [L], \ k \in [K].
\end{cases}
\tag{81}
$$

We have

$$\left\| W_{l,k}^{rt+v} - \bar{W}_l^{rt} \right\|_F \le \sum_{q=0}^{v-1} \left\| W_{l,k}^{rt+q+1} - W_{l,k}^{rt+q} \right\|_F \le \eta \left\| \sum_{q=0}^{v-1} \tilde{g}_{l,k}^{rt+q} \right\|_2 \overset{(i)}{\le} \eta \sum_{q=0}^{v-1} \left\| \tilde{g}_k^{rt+q} \right\|_2 \overset{(ii)}{\le} \eta\rho \sum_{q=0}^{v-1} \left\| g_k^{rt+q} \right\|_2 \tag{82}$$

$$\overset{(iii)}{\le} \frac{\eta\rho L}{N_k} \|X_k\|_F \left(\frac{7}{4}\right)^{L-1} \frac{\bar{\lambda}_{1\to L}}{\min_{l\in[L]} \bar{\lambda}_l} \sum_{q=0}^{v-1} \left\| f_{L,k}^{rt+q} - y_k \right\|_2 \tag{83}$$

$$\overset{(iv)}{\le} \frac{\eta\rho L}{N_k} \|X_k\|_F \left(\frac{7}{4}\right)^{L-1} \frac{\bar{\lambda}_{1\to L}}{\min_{l\in[L]} \bar{\lambda}_l} \sum_{q=0}^{v-1} (1+3\rho C'\eta)^q \left\| f_{L,k}^{rt} - y_k \right\|_2, \tag{84}$$

where (i) is because the norm of concentrated gradient is no smaller than norm of one-layer gradient; (ii) results from Lemma (1); (iii) comes from (31) in Lemma 4; (iv) is because of the induction assumption. Let $\eta < \frac{1}{\rho C'}$, we have

$$\left\| W_{l,k}^{rt+v} - \bar{W}_l^{rt} \right\|_F \le \frac{\eta\rho L}{N_k} \|X_k\|_F \left(\frac{7}{4}\right)^{L-1} \frac{\bar{\lambda}_{1\to L}}{\min_{l\in[L]} \bar{\lambda}_l} \sum_{q=0}^{v-1} (1+3\rho C'\eta)^q \left\| f_{L,k}^{rt} - y_k \right\|_2 \tag{85}$$

$$\overset{(i)}{\le} \frac{\eta\rho L}{N_k} \|X_k\|_F \left(\frac{7}{4}\right)^{l-1} \frac{\bar{\lambda}_{1\to L}}{\min_{l\in[L]} \bar{\lambda}_l} \sum_{q=0}^{v-1} 2^v \left\| f_{L,k}^{rt} - y_k \right\|_2$$

$$\overset{(ii)}{\le} \frac{\eta\rho L}{N_k} \|X_k\|_F \left(\frac{7}{4}\right)^{l-1} \frac{\bar{\lambda}_{1\to L}}{\min_{l\in[L]} \bar{\lambda}_l} \sum_{q=0}^{v-1} 2^v \left\| f_L^{rt} - y \right\|_2$$

$$\le \frac{\eta\rho L(2^r-1)}{N_k} \|X_k\|_F \left(\frac{7}{4}\right)^{L-1} \frac{\bar{\lambda}_{1\to L}}{\min_{l\in[L]} \bar{\lambda}_l} \| f_L^{rt} - y \|_2 \tag{86}$$

$$\le \frac{\eta\rho L(2^r-1)}{N_k} \|X_k\|_F \left(\frac{7}{4}\right)^{L-1} \frac{\bar{\lambda}_{1\to L}}{\min_{l\in[L]} \bar{\lambda}_l} \| f_L^0 - y \|_2 (1+3\rho C\eta)^{\frac{T}{2}}$$

$$\le \frac{\eta\rho L(2^r-1)}{N_k} \|X_k\|_F \left(\frac{7}{4}\right)^{L-1} \frac{\bar{\lambda}_{1\to L}}{\min_{l\in[L]} \bar{\lambda}_l} \| f_L^0 - y \|_2 \cdot \frac{\xi N\Phi\left(\theta^0\right)}{\epsilon}$$

$$\le \begin{cases} \frac{1}{4}\underline{\lambda}_l, & l \in \{3,\dots,L\}, \\ \frac{1}{6}, & l \in \{1,2\} \end{cases}.$$

where (i) uses $\eta < \frac{1}{\rho C'}$; (ii) is because $\|f_{L,k}^{rt} - y_k\|_2 \le \|f_L^{rt} - y\|_2$; the last inequality holds if we choose small enough $\eta$. To be more specific, we can choose

$$\eta < \frac{\min\left(\min_{l\in[L]} \frac{1}{4}\underline{\lambda}_l, \frac{1}{6}\right)}{\frac{\rho L(2^r-1)}{N_k} \|X_k\|_F \left(\frac{7}{4}\right)^{L-1} \frac{\bar{\lambda}_{1\to L}}{\min_{l\in[L]} \bar{\lambda}_l} \|f_L^0 - y\|_2 \cdot \frac{\xi\Phi(\theta^0)}{\epsilon}} \tag{87}$$

By Weyl's inequality, we have

$$\begin{cases} \sigma_{\min}\left(W_{l,k}^{rt+v+1}\right) \geqslant \sigma_{\min}\left(W_{l,k}^{rt}\right) - \frac{1}{4}\underline{\lambda}_l = \frac{1}{4}\underline{\lambda}_l, & l \in \{3,\dots,L\}, \ k \in [K], \\ \sigma_{\max}\left(W_{l,k}^{rt+v+1}\right) \leq \sigma_{\max}\left(W_{l,k}^{rt}\right) + \frac{1}{4}\bar{\lambda}_l \leqslant \frac{7}{4}\bar{\lambda}_l, & l \in \{3,\dots,L\}, \ k \in [K], \\ \sigma_{\max}\left(W_{1,k}^{rt}\right) \leq \frac{1}{6} + 1 + \|W_{1,k}^{rt}\|_2 \leq \frac{7}{4}\bar{\lambda}_l, & k \in [K], \\ \sigma_{\max}\left(W_{2,k}^{rt}\right) \leq \frac{1}{6} + 1 + \|W_{2,k}^{rt}\|_2 \leq \frac{7}{4}\bar{\lambda}_l, & k \in [K]. \end{cases} \tag{88}$$

(2) We next show that

$$\sigma_{\min}\left(F_{1,k}^{rt+q}\right) \geqslant \frac{1}{4}\underline{\alpha}_{0,k}, \ q \in \{0,\dots,v\}, \ k \in [K]. \tag{89}$$

It is sufficient to show $\sigma_{\min}\left(F_{1,k}^{rt+v}\right) \geqslant \frac{1}{4}\underline{\alpha}_{0,k}, \ k \in [K]$.

$$\left\|F_{1,k}^{rt+v} - F_{1,k}^{rt}\right\|_F = \left\|\sigma\left(X_k W_{1,k}^{rt+v}\right) - \sigma\left(X_k \bar{W}_{1,k)}^{rt}\right)\right\|_F \tag{90}$$

$$\overset{(i)}{\leq} \sigma_{\max}(X_k)\|W_{1,k}^{rt+v} - W_{1,k}^{rt}\|_F \tag{91}$$

$$\overset{(ii)}{\leq} \sigma_{\max}(X_k)\frac{\eta\rho(2^r - 1)}{N_k}\|X_k\|_F \left(\frac{7}{4}\right)^{L-1} \frac{\bar{\lambda}_{1\to L}}{\min_{l\in[L]} \bar{\lambda}_l}\|f_L^0 - y\|_2 \cdot \frac{\xi\Phi\left(\theta^0\right)}{\epsilon} \tag{92}$$

where (i) results from the Lipschitz gradient of $\sigma$ in Assumption 3 and (ii) comes from (86). If we choose small enough $\eta$, which satisfies

$$\eta < \frac{\frac{1}{4}\underline{\alpha}_{0,k}N_k}{\sigma_{\max}(X_k)\rho(2^r - 1)\|X_k\|_F \left(\frac{7}{4}\right)^{L-1} \frac{\bar{\lambda}_{1\to L}}{\min_{l\in[L]} \bar{\lambda}_l}\|f_{L,k}^0 - y\|_2 \cdot \frac{\xi\Phi(\theta^0)}{\epsilon}} \tag{93}$$

then we have

$$\left\|F_{1,k}^{rt+v} - F_{1,k}^{rt}\right\|_F \leq \frac{1}{4}\underline{\alpha}_{0,k} \tag{94}$$

(3) Next, we show that

$$\rho^{rt+v} \leq \rho. \tag{95}$$

Since we have already shown in (81) that $\sigma_{\max}(W_{l,k}^{rt+v}) \leq \frac{7}{4}\bar{\lambda}_l$, $\sigma_{\min}(W_{l,k}^{rt+v}) \geq \frac{1}{4}\bar{\lambda}_l$ and we have shown in (89) that $\sigma_{\min}(F_1^{rt+v}) \geq \frac{1}{4}\underline{\alpha}_{0,k}$. By lemma 1, we have

$$\rho(\theta_k^{rt+v}) \leq \frac{\left(\frac{7}{4}\right)^{L-1}LN\|X\|_F \frac{\bar{\lambda}_{1\to L}}{\min_{l\in[L]} \bar{\lambda}_l}}{\left(\frac{1}{4}\right)^{L-1}m\gamma^{L-2}\underline{\lambda}_{3\to L} \min_{k\in[K]} \underline{\alpha}_{0,k}} \leq \rho. \tag{96}$$

(4) Next, we prove

$$\Phi_k\left(\theta_k^{rt+q}\right) \leqslant (1 + 3\rho C'\eta)^q \Phi_k\left(\theta_k^{rt}\right), \ q \in \{0, \ldots, v\}, \quad k \in [K]. \tag{97}$$

We show

$$\Phi_k\left(\theta_k^{rt+v}\right) \leqslant (1 + 3\rho C'\eta)^v \Phi_k\left(\theta_k^{rt}\right). \tag{98}$$

First, we need to show $\Phi_k$ has Lipschitz gradient within $[\theta^{rt+v-1}, \theta^{rt+v}]$. This is similar to the proof of (50) in Lemma 8. So we don't include the details here. It is easy to show that, for $\theta_k^{rt+v-1,s} := \theta_k^{rt+v-1} + s(\theta_k^{rt+v} - \theta_k^{rt+v-1})$, there is $\max\left(\sigma_{\max}\left(W_{l,k}^{rt+v-1,s}\right), \sigma_{\max}\left(W_{l,k}^{rt+v-1}\right)\right) \leq \frac{7}{4}\bar{\lambda}_l$. So similarly we can derive the Lipschitz constant

$$Q_k = \frac{L\sqrt{L}}{N_k}\left(\frac{7}{4}\right)^{2(L-1)}\|X_k\|_F^2 \frac{\bar{\lambda}_{1\to L}^2}{\min_{l\in[L]} \bar{\lambda}_l^2} + \frac{L\sqrt{L}}{N_k}\|X_k\|_F \left(1 + L\beta\|X_k\|_F R'\right) R' \left\|f_{L,k}^0 - y_k\right\|_2, \tag{99}$$

such that $\forall s \in [0,1]$,

$$\left\|g_k^{rt+v-1,s} - g_k^{rt+v-1}\right\|_2 \leq Q_k\|\theta_k^{rt+v-1,s} - \theta_k^{rt+v-1}\|_2. \tag{100}$$

With Lipschitz gradient within $[\theta_k^{rt+v-1}, \theta_k^{rt+v}]$, by Lemma 5, we have

$$\Phi_k\left(\theta_k^{rt+v}\right) \leq \Phi_k\left(\theta_k^{rt+v-1}\right) + \left\langle \nabla\Phi_k\left(\theta_k^{rt+v-1}\right), \theta_k^{rt+v} - \theta_k^{rt+v}\right\rangle + \frac{Q_k}{2}\left\|\theta_k^{rt+v-1} - \theta_k^{rt+v-1}\right\|_2^2 \tag{101}$$

$$= \Phi_k\left(\theta_k^{rt+v-1}\right) + \left\langle g_k^{rt+v-1}, -\eta\tilde{g}_k^{rt+v-1}\right\rangle + \frac{Q_k}{2}\left\|\eta\tilde{g}_k^{rt+v-1}\right\|_2^2 \tag{102}$$

$$\leq \Phi_k\left(\theta_k^{rt+v-1}\right) + \eta\left\|g_k^{rt+v-1}\right\|_2\left\|\tilde{g}_k^{rt+v-1}\right\|_2 + \frac{Q_k}{2}\eta^2\left\|\tilde{g}_k^{rt+v-1}\right\|_2^2 \tag{103}$$

$$\leq \Phi_k\left(\theta_k^{rt+v-1}\right) + \eta\rho\left\|g_k^{rt+v-1}\right\|_2^2 + \frac{Q_k}{2}\eta^2\rho^2\left\|g_k^{rt+v-1}\right\|_2^2 \tag{104}$$

Let $\eta < \frac{1}{Q_k \rho}$, we have the above inequality

$$\Phi_k\left(\theta_k^{rt+v}\right) \leq \Phi_k\left(\theta_k^{rt+v-1}\right) + \eta\rho\left\|g_k^{rt+v-1}\right\|_2^2 + \frac{Q_k}{2}\eta^2\rho^2\left\|g_k^{rt+v-1}\right\|_2^2 \tag{105}$$

$$\leq \Phi_k\left(\theta_k^{rt+v-1}\right) + \frac{3}{2}\rho\eta\left\|g_k^{rt+v-1}\right\|_2^2 \tag{106}$$

$$\leq \Phi_k\left(\theta_k^{rt+v}\right) + \frac{3\rho\eta L}{N_k}\left(\frac{7}{4}\right)^{2(L-1)}\frac{\bar{\lambda}_{1\to L}^2}{\min\limits_{l\in[L]}\bar{\lambda}_l^2}\Phi_k\left(\theta_k^{rt+v-1}\right), \tag{107}$$

where the third inequality comes from (31) in Lemma 4. Recall $C' := \max\limits_k(\frac{1}{N_k}\left(\frac{7}{4}\right)^{2(L-1)}\frac{\bar{\lambda}_{1\to L}^2}{\min\limits_{l\in[L]}\bar{\lambda}_l^2})$, we have

$$\Phi_k\left(\theta_k^{rt+v+1}\right) \leq \Phi_k\left(\theta_N^{rt+v}\right)(1+3\rho C'\eta). \tag{108}$$

Now Step 1 is proved. Next we show Step 2.
(1) Show

$$\begin{cases} \sigma_{\max}\left(\bar{W}_l^{ru}\right) \leqslant \frac{3}{2}\bar{\lambda}_l, \ u \in \{0,1,\ldots t+1\}, \ l \in [L] \\ \sigma_{\min}\left(\bar{W}_l^{ru}\right) \geqslant \frac{1}{2}\bar{\lambda}_l \quad u \in \{0,1,\ldots t+1\} \quad l \in \{3,\ldots,L\} \end{cases}. \tag{109}$$

Define $\tilde{\nabla}_{W_l,k}\Phi_k(\theta_k^{rt})$ be the stochastic gradient over layer $l$ of each client. Denote $\bar{\tilde{g}}^{l,rt+v} := \sum\limits_{k=1}^K\frac{N_k}{N}\tilde{\nabla}_{W_l,k}\Phi_k(\theta_k^{rt+v})$ We have

$$\left\|\bar{W}_l^{r(t+1)} - W_l^0\right\|_F = \eta\sum_{u=0}^t\left\|\bar{\tilde{g}}^{l,ru} + \bar{\tilde{g}}^{l,ru+1} + \ldots + \bar{\tilde{g}}^{l,rt+r-1}\right\|_2 \tag{110}$$

$$\leq \eta\sum_{u=0}^t\sum_{v=0}^{r-1}\left\|\bar{\tilde{g}}^{l,ru+v}\right\|_2 \tag{111}$$

$$\leq \eta\sum_{u=0}^t\sum_{v=0}^{r-1}\left\|\bar{\tilde{g}}^{ru+v}\right\|_2 \tag{112}$$

By Step 1, we know for $v \in \{0,1,\ldots,r-1\}$, we have $\rho^{rt+v} \leq \rho$. So by definition of $\rho^{rt+v}$, we have $\|\tilde{g}^{ru+v}\|_2 \leq \rho\|g^{ru+v}\|_2$. Then it is easy to verify that the assumptions in Lemma 8 are satisfied, where $\Lambda_l = \frac{7}{4}\bar{\lambda}_l$, $Q_k$ is defined in (63) and $A = 1+3\rho C'\eta$. Then by Lemma 8, if $\eta < \frac{1}{\rho C'}$, we have

$$\eta\sum_{u=0}^t\sum_{v=0}^{r-1}\left\|\bar{\tilde{g}}^{ru+v}\right\|_2 \leq \eta\sum_{u=0}^t\frac{\rho L\|X\|_F}{N}\left(\frac{7}{4}\right)^{L-1}(2^r-1)\frac{\bar{\lambda}_{1\to L}}{\min\limits_{l\in[L]}\bar{\lambda}_l}\|f_L^{rt} - y\|_2 \tag{113}$$

Using the definition of $P = \frac{L\|X\|_F}{N}\left(\frac{7}{4}\right)^{L-1}(2^r-1)$, we have

$$\eta\sum_{u=0}^t\sum_{v=0}^{r-1}\left\|\bar{\tilde{g}}^{ru+v}\right\|_2 \leq \eta\sum_{u=0}^t\frac{\rho L\|X\|_F}{N}\left(\frac{7}{4}\right)^{L-1}(2^r-1)\frac{\bar{\lambda}_{1\to L}}{\min\limits_{l\in[L]}\bar{\lambda}_l}\|f_L^{rt} - y\|_2 \tag{114}$$

$$\leq \eta\rho P\frac{\bar{\lambda}_{1\to L}}{\min\limits_{l\in[L]}\bar{\lambda}_l}\sum_{u=0}^t\|f_L^{rt} - y\|_2 \tag{115}$$

$$\leq \eta\rho P\frac{\bar{\lambda}_{1\to L}}{\min\limits_{l\in[L]}\bar{\lambda}_l}\sum_{u=0}^t(1+3\rho C\eta)^{\frac{u}{2}}\|f_L^0 - y\|_2, \tag{116}$$

$$\tag{117}$$

where the last inequality comes from the induction assumption. Now let $S = \sqrt{1 + 3\rho C\eta}$, if we choose $\eta < \frac{1}{\rho C}$, we get

$$
\begin{aligned}
\eta \sum_{u=0}^{t} \sum_{v=0}^{r-1} \|\bar{\tilde{g}}^{ru+v}\|_2 &\leq \eta \rho P \frac{\bar{\lambda}_{1 \to L}}{\min_{l \in [L]} \bar{\bar{\lambda}}_l} \sum_{u=0}^{t} (1 + 3\rho C\eta)^{\frac{u}{2}} \|f_L^0 - y\|_2 \\
&= \eta \rho P \frac{\bar{\lambda}_{1 \to L}}{\min_{l \in [L]} \bar{\bar{\lambda}}_l} \sum_{u=0}^{t} S^u \|f_L^0 - y\|_2 \\
&\leq \eta \rho P \frac{\bar{\lambda}_{1 \to L}}{\min_{l \in [L]} \bar{\bar{\lambda}}_l} \frac{S^{T+1}}{S^2 - 1} (S+1) \|f_L^0 - y\|_2 \\
&= \eta \rho P \frac{\bar{\lambda}_{1 \to L}}{\min_{l \in [L]} \bar{\bar{\lambda}}_l} \frac{S^{T+1} \cdot 3}{3\rho C\eta} \|f_L^0 - y\|_2
\end{aligned}
\tag{118}
$$

By Lemma 6, we have $S^T \leq \frac{\xi \Phi(\theta^0)}{\epsilon}$. Additionally, $S \leq 2$, therefore, we have

$$
\begin{aligned}
\eta \sum_{u=0}^{t} \sum_{v=0}^{r-1} \|\bar{\tilde{g}}^{ru+v}\|_2 &\leq \eta \rho P \frac{\bar{\lambda}_{1 \to L}}{\min_{l \in [L]} \bar{\bar{\lambda}}_l} \frac{2S^T \cdot 3}{3\rho C\eta} \|f_L^0 - y\|_2
\tag{119} \\
&\leq \frac{P}{C} \frac{\bar{\lambda}_{1 \to L}}{\min_{l \in [L]} \bar{\bar{\lambda}}_l} \cdot \frac{2\xi \Phi(\theta^0)}{\epsilon} \|f_L^0 - y\|_2 \\
&= \frac{2}{L\|X\|_F \left(\frac{3}{2}\right)^{L-1} \frac{\bar{\lambda}_{1 \to L}}{\min_{l \in [L]} \bar{\lambda}_l}} \cdot \frac{\xi \Phi(\theta^0)}{\epsilon} \|f_L^0 - y\|_2
\tag{120} \\
&\leq \begin{cases} \frac{1}{2}\underline{\lambda}_l, & l \in \{3, \dots, L\}, \\ 1, & l \in \{1, 2\}. \end{cases}
\end{aligned}
$$

where the last inequality is from (68). So by Weyl's ineuality, we have

$$
\begin{cases}
\sigma_{\min}\left(\bar{W}_l^{r(t+1)}\right) \geqslant \sigma_{\min}\left(\bar{W}_l^{rt}\right) - \frac{1}{2}\underline{\lambda}_l = \frac{1}{2}\underline{\lambda}_l, & l \in \{3, \dots, L\}, \ k \in [K], \\
\sigma_{\max}\left(\bar{W}_l^{r(t+1)}\right) \leq \sigma_{\max}\left(\bar{W}_l^{rt}\right) + \frac{1}{2}\bar{\lambda}_l \leqslant \frac{3}{2}\bar{\lambda}_l, & l \in \{3, \dots, L\}, \ k \in [K], \\
\sigma_{\max}\left(\bar{W}_1^{rt}\right) \leq 1 + \sigma_{\max}\left(\bar{W}_1^{rt}\right) \leq \frac{3}{2}\bar{\lambda}_l, & k \in [K], \\
\sigma_{\max}\left(\bar{W}_2^{rt}\right) \leq 1 + \sigma_{\max}(\bar{W}_{2,k}^{rt}) \leq \frac{3}{2}\bar{\lambda}_l, & k \in [K].
\end{cases}
\tag{121}
$$

(2) Show

$$
\sigma_{\min}\left(F_1^{ru}\right) \geqslant \frac{1}{2}\underline{\alpha}_0, \ u \in \{0, \dots, t+1\} \quad l \in [L].
\tag{122}
$$

Similarly, we have

$$
\left\|F_1^{r(t+1)} - F_1^0\right\|_F = \left\|\sigma\left(X\bar{W}_1^{r(t+1)}\right) - \sigma\left(XW_1^0\right)\right\|_F
\tag{123}
$$

$$
\overset{(i)}{\leq} \sigma_{\max}(X) \left\|\bar{W}_1^{r(t+1)} - W_1^0\right\|_F
\tag{124}
$$

$$
\overset{(ii)}{\leq} \sigma_{\max}(X) \frac{2}{L\|X\|_F \left(\frac{3}{2}\right)^{L-1} \frac{\bar{\lambda}_{1 \to L}}{\min_{l \in [L]} \bar{\lambda}_l}} \cdot \frac{\xi \Phi(\theta^0)}{\epsilon} \|f_L^0 - y\|_2
\tag{125}
$$

$$
\leq \|X\|_F \frac{2}{L\|X\|_F \left(\frac{3}{2}\right)^{L-1} \frac{\bar{\lambda}_{1 \to L}}{\min_{l \in [L]} \bar{\lambda}_l}} \cdot \frac{\xi \Phi(\theta^0)}{\epsilon} \|f_L^0 - y\|_2
\tag{126}
$$

$$
\overset{(iii)}{\leq} \frac{1}{2}\underline{\alpha}_0,
\tag{127}
$$

where (i) is because $\sigma$ is $1-$Lipschitz; (ii) comes from (120), and (iii) is because (69) in B.2. So similarly by Weyl's inequality, we have $\sigma_{\min}\left(F_1^{r(t+1)}\right) \geq \sigma_{\min}\left(F_1^0\right) = \underline{\alpha}_0 - \frac{1}{2}\underline{\alpha}_0 = \frac{1}{2}\underline{\alpha}_0$.

(3) Show

$$\sigma_{\min}\left(F_{1,k}^{ru}\right) \geqslant \frac{1}{2}\underline{\alpha}_{0,k}, \; u \in \{0,\ldots,t+1\} \quad l \in [L]. \tag{128}$$

Similarly, we have

$$\left\|F_{1,k}^{r(t+1)} - F_{1,k}^0\right\|_F = \left\|\sigma\left(X_k \bar{W}_1^{r(t+1)}\right) - \sigma\left(X_k W_1^0\right)\right\|_F \tag{129}$$

$$\overset{(i)}{\leq} \sigma_{\max}(X_k)\left\|\bar{W}_1^{r(t+1)} - W_1^0\right\|_F \tag{130}$$

$$\overset{(ii)}{\leq} \sigma_{\max}(X_k)\frac{2}{L\|X\|_F \left(\frac{3}{2}\right)^{L-1} \frac{\bar{\lambda}_{1 \to L}}{\underset{l \in [L]}{\min} \bar{\lambda}_l}} \cdot \frac{\xi\Phi\left(\theta^0\right)}{\epsilon}\|f_L^0 - y\|_2 \tag{131}$$

$$\leq \|X_k\|_F\frac{2}{L\|X\|_F \left(\frac{3}{2}\right)^{L-1} \frac{\bar{\lambda}_{1 \to L}}{\underset{l \in [L]}{\min} \bar{\lambda}_l}} \cdot \frac{\xi\Phi\left(\theta^0\right)}{\epsilon}\|f_L^0 - y\|_2 \tag{132}$$

$$\overset{(iii)}{\leq} \frac{1}{2}\underline{\alpha}_{0,k}, \tag{133}$$

where (i) is because $\sigma$ is $1-$Lipschitz; (ii) comes from (120), and (iii) is because (69) in B.2. So similarly by Weyl's inequality, we have $\sigma_{\min}\left(F_{1,k}^{r(t+1)}\right) \geq \sigma_{\min}\left(F_{1,k}^0\right) = \underline{\alpha}_{0,k} - \frac{1}{2}\underline{\alpha}_{0,k} = \frac{1}{2}\underline{\alpha}_{0,k}$.

(3) Show

$$\rho^{r(t+1)} \leq \rho. \tag{134}$$

Since we have already shown in (81) that $\sigma_{\max}\left(W_{l,k}^{r(t+1)}\right) \leq \frac{3}{2}\bar{\lambda}_l$, $\sigma_{\min}\left(W_{l,k}^{r(t+1)}\right) \geq \frac{1}{2}\bar{\lambda}_l$, and we have shown in (89) that $\sigma_{\min}(F_1^{r(t+1)}) \geq \frac{1}{2}\underline{\alpha}_0$. By Lemma 1, we have

$$\rho^{r(t+1)} \leq \frac{\left(\frac{3}{2}\right)^{L-1} LN \frac{\bar{\lambda}_{1 \to L}}{\underset{l \in [L]}{\min} \bar{\lambda}_l}}{\left(\frac{1}{2}\right)^{L-1} m\gamma^{L-2}\underline{\lambda}_{3 \to L} \underset{k \in [K]}{\min} \underline{\alpha}_0} < \rho. \tag{135}$$

(4) Show

$$\Phi\left(\bar{\theta}^{ru}\right) \leqslant (1 + 3\rho C\eta)^u \Phi\left(\theta^0\right), \; u \in \{0,\ldots,t+1\}, \; l \in [L].$$

First, similar to the proof of (50), we can derive

$$Q = \frac{L\sqrt{L}}{N} \cdot \left(\frac{3}{2}\right)^{2(L-1)} \|X\|_F^2 \frac{\bar{\lambda}_{1 \to L}^2}{\underset{l \in [L]}{\min} \bar{\lambda}_l^2} + \frac{L\sqrt{L}}{N}\|X\|_F \left(1 + L\beta\|X\|_F R\right) R\|f_L^0 - y\|_2, \tag{136}$$

where $R = \prod_{p=1}^{L} \max\left(1, \frac{3}{2}\bar{\lambda}_p\right)$, such that $\forall \bar{\theta}^{rt,s} = \bar{\theta}^{rt} + s(\bar{\theta}^{r(t+1)} - \bar{\theta}^{rt}), s \in [0,1]$, we have

$$\left\|g^{r(t+1),s} - g^{r(t+1)}\right\|_2 \leq Q\left\|\bar{\theta}^{r(t+1),s} - \bar{\theta}^{rt}\right\|_2. \tag{137}$$

Then by Lemma 5 we have

$$\Phi\left(\bar{\theta}^{r(t+1)}\right) = \Phi\left(\bar{\theta}^{rt} - \eta\bar{\tilde{g}}^{rt} - \ldots - \eta\bar{\tilde{g}}^{rt+r-1}\right)$$

$$\leq \Phi\left(\bar{\theta}^{rt}\right) - \eta\left\langle g^{rt}, \bar{\tilde{g}}^{rt} + \ldots + \bar{\tilde{g}}^{rt+r-1}\right\rangle + \frac{Q}{2}\eta^2\left\|\bar{\tilde{g}}^{rt} + \ldots + \bar{\tilde{g}}^{rt+r-1}\right\|_2^2 \tag{138}$$

$$\leq \Phi\left(\bar{\theta}^{rt}\right) + \eta\left\|g^{rt}\right\|_2\left\|\bar{\tilde{g}}^{rt} + \ldots + \bar{\tilde{g}}^{rt+r-1}\right\|_2 + \frac{Q}{2}\eta^2\left\|\bar{\tilde{g}}^{rt} + \ldots + \bar{\tilde{g}}^{rt+r-1}\right\|_2^2$$

By (49) in Lemma 8, if $\eta < \frac{1}{\rho C}$, we have $A = 2$, and we have

$$\left\| \bar{\bar{g}}^{rt} + \ldots + \bar{\bar{g}}^{rt+r-1} \right\|_2 \leq \frac{\rho L \|X\|_F}{N} \left(\frac{7}{4}\right)^{L-1} (2^r - 1) \frac{\bar{\lambda}_{1\to L}}{\min\limits_{l\in[L]} \bar{\lambda}_l} \|f_L^{rt} - y\|_2 = \rho P \frac{\bar{\lambda}_{1\to L}}{\min\limits_{l\in[L]} \bar{\lambda}_l} \|f_L^{rt} - y\|_2.$$

$$(139)$$

Then we have

$$\Phi\left(\bar{\theta}^{r(t+1)}\right) \leq \Phi\left(\bar{\theta}^{rt}\right) + \eta \left\|g^{rt}\right\|_2 \left\|\bar{\bar{g}}^{rt} + \ldots + \bar{\bar{g}}^{rt+r-1}\right\|_2 + \frac{Q}{2}\eta^2 \left\|\bar{\bar{g}}^{rt} + \ldots + \bar{\bar{g}}^{rt+r-1}\right\|_2^2$$

$$(140)$$

$$\leq \Phi\left(\bar{\theta}^{rt}\right) + \eta \left\|g^{rt}\right\|_2 \cdot \rho P \frac{\bar{\lambda}_{1\to L}}{\min\limits_{l\in[L]} \bar{\lambda}_l} \|f_L^{rt} - y\|_2 + \frac{Q}{2}\eta^2 \rho^2 P^2 \frac{\bar{\lambda}_{1\to L}^2}{\min\limits_{l\in[L]} \bar{\lambda}_l^2} \|f_L^{rt} - y\|_2^2$$

$$(141)$$

$$\leq \Phi\left(\bar{\theta}^{rt}\right) + \eta \left\|g^{rt}\right\|_2 \cdot \rho P \frac{\bar{\lambda}_{1\to L}}{\min\limits_{l\in[L]} \bar{\lambda}_l} \|f_L^{rt} - y\|_2 + \frac{Q}{2}\eta^2 \rho^2 P^2 \frac{\bar{\lambda}_{1\to L}^2}{\min\limits_{l\in[L]} \bar{\lambda}_l^2} \|f_L^{rt} - y\|_2^2$$

$$(142)$$

$$\overset{(i)}{\leq} \Phi\left(\bar{\theta}^{rt}\right) + \frac{\eta \rho P L \|X\|_F}{N} \left(\frac{3}{2}\right)^{L-1} \frac{\bar{\lambda}_{1\to L}^2}{\min\limits_{l\in[L]} \bar{\lambda}_l^2} \|f_L^{rt} - y\|_2^2 + \frac{Q}{2}\eta^2 \rho^2 P^2 \frac{\bar{\lambda}_{1\to L}^2}{\min\limits_{l\in[L]} \bar{\lambda}_l^2} \|f_L^{rt} - y\|_2^2,$$

$$(143)$$

where (i) comes from (31) in Lemma 4. Let $\eta < \frac{L\|X\|_F \left(\frac{3}{2}\right)^{L-1}}{Q\rho P N} = \frac{\left(\frac{6}{7}\right)^{L-1}}{Q\rho(2^r-1)}$, we get

$$\Phi\left(\bar{\theta}^{r(t+1)}\right) \leq \Phi\left(\bar{\theta}^{rt}\right) + \frac{\eta \rho P L \|X\|_F}{N} \left(\frac{3}{2}\right)^{L-1} \frac{\bar{\lambda}_{1\to L}^2}{\min\limits_{l\in[L]} \bar{\lambda}_l^2} \|f_L^{rt} - y\|_2^2 + \frac{Q}{2}\eta^2 \rho^2 P^2 \frac{\bar{\lambda}_{1\to L}^2}{\min\limits_{l\in[L]} \bar{\lambda}_l^2} \|f_L^{rt} - y\|_2^2$$

$$(144)$$

$$\leq \Phi\left(\bar{\theta}^{rt}\right) + \frac{3}{2} \cdot \frac{\eta \rho P L \|X\|_F}{N} \left(\frac{3}{2}\right)^{L-1} \frac{\bar{\lambda}_{1\to L}^2}{\min\limits_{l\in[L]} \bar{\lambda}_l^2} \|f_L^{rt} - y\|_2^2$$

$$\leq \Phi\left(\bar{\theta}^{rt}\right) \left(1 + 3\frac{\eta \rho P L \|X\|_F}{N} \left(\frac{3}{2}\right)^{L-1} \frac{\bar{\lambda}_{1\to L}^2}{\min\limits_{l\in[L]} \bar{\lambda}_l^2}\right)$$

Recall $C = P L \|X\|_F \left(\frac{3}{2}\right)^{L-1} \frac{\bar{\lambda}_{1\to L}^2}{\min\limits_{l\in[L]} \bar{\lambda}_l^2}$, then we have

$$\Phi\left(\bar{\theta}^{r(t+1)}\right) \leq (1 + 3\rho C \eta) \Phi\left(\bar{\theta}^{rt}\right). \tag{145}$$

(5) Show

$$E\left[\Phi(\bar{\theta}^{r(t+1)})\right] \leq (1 - \mu C \eta)^u \Phi\left(\theta^0\right), \; u \in \{0, 1, \ldots, t+1\}$$

By (138), we have

$$\Phi(\bar{\theta}^{r(t+1)}) \leq \Phi\left(\bar{\theta}^{rt}\right) - \eta \left\langle g^{rt}, \tilde{g}^{rt} + \ldots + \tilde{g}^{rt+r-1} \right\rangle + \frac{Q}{2}\eta^2 \left\|\bar{\bar{g}}^{rt} + \ldots + \bar{\bar{g}}^{rt+r-1}\right\|_2^2 \tag{146}$$

Given $\bar{\theta}^{rt}$, take expectation of the stochastic gradient on both sides conditioned on $\bar{\theta}^{rt}$ and the past, we get

$$
\begin{aligned}
E[\Phi(\bar{\theta}^{r(t+1)})] &\leq E\Big[\Phi\left(\bar{\theta}^{rt}\right) - \eta\left\langle g^{rt}, \bar{g}^{rt} + \ldots + \bar{g}^{rt+r-1}\right\rangle + \frac{Q}{2}\eta^2\rho^2 P^2 \frac{\bar{\lambda}_{1\to L}^2}{\min\limits_{l\in[L]}\bar{\lambda}_l^2}\|f_L^{rt} - y\|_2^2\Big] \\
&\leq E\Big[\Phi\left(\bar{\theta}^{rt}\right) - \eta\left\langle g^{rt}, r\bar{g}^{rt}\right\rangle - \eta\langle g^{rt}, \sum_{v=1}^{r-1}\bar{g}^{rt+v} - \bar{g}^{rt}\rangle + \frac{Q}{2}\eta^2\rho^2 P^2 \frac{\bar{\lambda}_{1\to L}^2}{\min\limits_{l\in[L]}\bar{\lambda}_l^2}\|f_L^{rt} - y\|_2^2\Big] \\
&\leq E\Big[\Phi\left(\bar{\theta}^{rt}\right) - \eta r\|g^{rt}\|_2^2 + \eta\|g^{rt}\|_2 \times \|\sum_{v=1}^{r-1}\bar{g}^{rt+v} - \bar{g}^{rt}\|_2 + \frac{Q}{2}\eta^2\rho^2 P^2 \frac{\bar{\lambda}_{1\to L}^2}{\min\limits_{l\in[L]}\bar{\lambda}_l^2}\|f_L^{rt} - y\|_2^2\Big]
\end{aligned}
\tag{147}
$$

Now it is easy to verify the assumptions in Lemma 1 are satisfied. Let $\bar{\Lambda}_l = \frac{7}{4}\bar{\lambda}_l$, $Q$ defined in (136), $A = 1 + 3\rho C'\eta \leq 2$, by (51) in Lemma 8:, we have

$$
\left\|\sum_{v=1}^{r-1}\bar{g}^{rt+v} - \bar{g}^{rt}\right\|_2 \leq \frac{\eta\rho L(2^r - 1)}{N}\left(\frac{7}{4}\right)^{L-1}\frac{\bar{\lambda}_{1\to L}}{\min\limits_{l\in[L]}\bar{\lambda}_l}\sqrt{\sum_{k=1}^{K}Q_k^2\|X_k\|_F^2}\|f_L^{rt} - y\|_2
\tag{148}
$$

Plug (148) into (147), we get:

$$
\begin{aligned}
E\left[\Phi(\bar{\theta}^{r(t+1)})\right] &\leq E\Big[\Phi\left(\bar{\theta}^{rt}\right) - \eta r\|g^{rt}\|_2^2 + \eta\|g^{rt}\|_2 \times \|\sum_{v=1}^{r-1}\bar{g}^{rt+v} - \bar{g}^{rt}\|_2 + \frac{Q}{2}\eta^2\rho^2 P^2 \frac{\bar{\lambda}_{1\to L}^2}{\min\limits_{l\in[L]}\bar{\lambda}_l^2}\|f_L^{rt} - y\|_2^2\Big] \\
&\leq E\Big[\Phi\left(\bar{\theta}^{rt}\right) - \eta r\|g^{rt}\|_2^2 + \eta\|g^{rt}\|_2 \times \frac{\eta\rho L(2^r-1)\|X\|_F}{N}\left(\frac{7}{4}\right)^{L-1}\sqrt{\sum_{k=1}^{K}Q_k^2\|X_k\|_F^2} \\
&\quad \times \frac{\bar{\lambda}_{1\to L}}{\min\limits_{l\in[L]}\bar{\lambda}_l}\|f_L^{rt} - y\|_2 + \frac{Q}{2}\eta^2\rho^2 P^2 \frac{\bar{\lambda}_{1\to L}^2}{\min\limits_{l\in[L]}\bar{\lambda}_l^2}\|f_L^{rt} - y\|_2^2\Big] \\
&\stackrel{(i)}{\leq} E\Big[\Phi\left(\bar{\theta}^{rt}\right) - \eta r\|g^{rt}\|_2^2 + \frac{\eta^2 L^2\rho(2^r-1)\|X\|_F}{N^2} \times (\frac{21}{8})^{L-1}\sqrt{\sum_{k=1}^{K}Q_k^2\|X_k\|_F^2} \\
&\quad \times \frac{\bar{\lambda}_{1\to L}^2}{\min\limits_{l\in[L]}\bar{\lambda}_l^2}\|f_L^{rt} - y\|_2^2 + \frac{Q}{2}\rho^2\eta^2 P^2 \frac{\bar{\lambda}_{1\to L}^2}{\min\limits_{l\in[L]}\bar{\lambda}_l^2}\|f_L^{rt} - y\|_2^2\Big] \\
&\stackrel{(ii)}{\leq} E\Big[\Phi\left(\bar{\theta}^{rt}\right) - \eta\underbrace{\frac{r}{N^2}\gamma^{2(L-2)}\left(\frac{1}{2}\right)^{2(L-1)}\underline{\lambda}_{3\to L}^2\alpha_0^2}_{:=\mu'}\|f_L^{rt} - y\|_2^2 + \\
&\quad \eta^2\underbrace{\left(\frac{L^2\rho(2^r-1)\|X\|_F(\frac{21}{8})^{L-1}\sqrt{\sum_{k=1}^{K}Q_k^2\|X_k\|_F^2}}{N^2} + \frac{Q}{2}\rho^2 P^2\right)\frac{\bar{\lambda}_{1\to L}^2}{\min\limits_{l\in[L]}\bar{\lambda}_l^2}}_{:=B}\|f_L^{rt} - y\|_2^2\Big],
\end{aligned}
$$

where (i) uses (31) to provide an upperbound for $\|g^{rt}\|$, (ii) uses (28) to provide a lower bound for $\|g^{rt}\|$. Let $\eta < \frac{\mu'}{2B}$, we have

$$
E[\Phi(\bar{\theta}^{r(t+1)})] \leq E\big[\Phi\left(\bar{\theta}^{rt}\right) - \eta\mu'\|f_L^{rt} - y\|_2^2 + \eta^2 B\|f_L^{rt} - y\|_2^2\big]
\tag{149}
$$

$$
\leq E\big[\Phi\left(\bar{\theta}^{rt}\right)\left(1 - \eta\mu'\right)\big]
\tag{150}
$$

$$
= E[\Phi\left(\bar{\theta}^{rt}\right)]\left(1 - \eta\frac{r}{N}\gamma^{2(L-2)}\left(\frac{1}{2}\right)^{2(L-1)}\underline{\lambda}_{3\to L}^2\alpha_0^2\right)
\tag{151}
$$

Let $\mu = \frac{\frac{r}{N}\gamma^{2(L-2)}\left(\frac{1}{2}\right)^{2(L-1)}\underline{\lambda}_{3\to L}^2\underline{\alpha}_0^2}{C}$, we have

$$E\left[\Phi(\bar{\theta}^{r(t+1)})\right] \leq (1 - \mu C\eta)E[\Phi\left(\bar{\theta}^{rt}\right)], \tag{152}$$

where $\mu C = \mu'$. $\qquad\square$

*Now we summarize the choice of $\eta$, it should be smaller than all the following quantities:*

$$\frac{1}{\rho C'}, \frac{1}{\rho \max(Q_k)}, \frac{1}{\rho C}, \frac{1}{\mu C}, \frac{\min\left\{\min_{l\in[L]}\frac{1}{4}\underline{\lambda}_l, \frac{1}{6}\right\}}{\frac{\rho L(2^r-1)}{N_k}\|X_k\|_F\left(\frac{7}{4}\right)^{L-1}\frac{\bar{\lambda}_{1\to L}}{\min_{l\in[L]}\bar{\lambda}_l}\|f_L^0 - y\|_2 \cdot \frac{\xi\Phi(\theta^0)}{\epsilon}},$$

$$\frac{\frac{1}{4}\underline{\alpha}_{0,k}N_k}{\sigma_{\max}\left(X_k\right)\rho(2^r-1)\|X_k\|_F\left(\frac{7}{4}\right)^{L-1}\frac{\bar{\lambda}_{1\to L}}{\min_{l\in[L]}\bar{\lambda}_l}\left\|f_{L,k}^0 - y\right\|_2 \cdot \frac{\xi\Phi(\theta^0)}{\epsilon}}. \tag{153}$$

**Remark 8.** *To satisfy the initialization assumptions defined in (69) and (68), we initialize the neural network coefficients such that we have $\bar{\lambda}_{1\to L} \sim \mathcal{O}(1/\epsilon)$. Note from the definition of $\mu'$ in (152) that this implies $\mu' \sim \mathcal{O}(\underline{\lambda}_{3\to L}^2) = \mathcal{O}(\bar{\lambda}_{1\to L}^2) = \mathcal{O}(1/\epsilon^2)$. Also, note from the choice of step-size in (153) that, we have $\eta \sim \mathcal{O}(\epsilon \times 1/\bar{\lambda}_{1\to L}) = \mathcal{O}(\epsilon^2)$. Note that this follows from the fact that $\eta$ is smaller than each quantity defined in (153) above. Thus, we have $\mu'\eta = \mathcal{O}(1)$ and we can always choose $\eta = c/\mu'$ for some $c \in (0,1)$, which guarantees linear convergence of the objective in each communication round (see Theorem 1).*

## C  EXPERIMENT SETTING AND RESULT

### C.1  MODEL AND PARAMETER SETTINGS:

*To analyze the performance of FedAvg-SGD on the MNIST data set, we use a single hidden-layer fully-connected neural network (MLP) with ReLU activation. We set the hidden-layer size to be 32 (resp. 1,000) for the small (resp. large) network. We choose the mini-batch size $m = 10$ and choose the number of local steps to be $r = 10$. Using the above network, we also compare the random initialization with the special initialization strategy in (23),(24) with MNIST and Fashion MNIST dataset. For the CIFAR-10 data set, we analyze the performance of FedAvg-SGD on two network architectures – convolutional neural network (CNN) and ResNet. We design the smaller CNN using two $5 \times 5$ convolutional layers followed by $2 \times 2$ max pooling, each has 6 and 16 channels, connected by 2 fully-connected layers with 120 and 84 hidden neurons. For larger CNN, we use three $3 \times 3$ convolutional layers each with 128 channels followed by $2 \times 2$ max pooling. The ReLU activation function is used after each hidden layer for small/large CNN. For ResNet, we compare the performance on ResNet18 with ResNet50 architectures. For both the CNN and ResNet, we use a mini-batch size of $m = 32$ and number of local steps to be $r = 5$. We randomly sample 10 clients in each epoch and perform FedAvg-SGD for more efficient training.*

### C.2  EXPERIMENT RESULT FOR MNIST

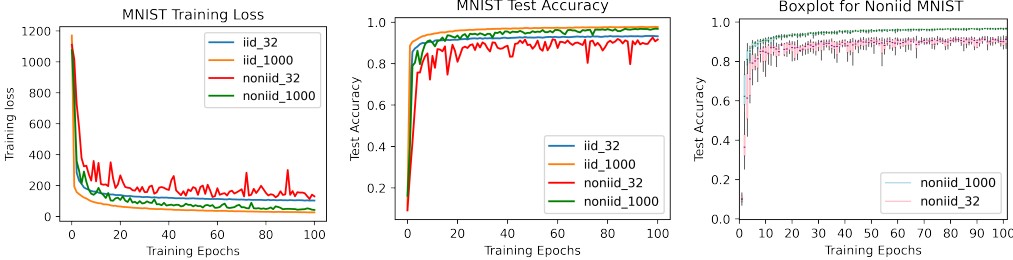

Figure 4: MNIST with MLP: Comparison of FedAvg on large and small size MLP.

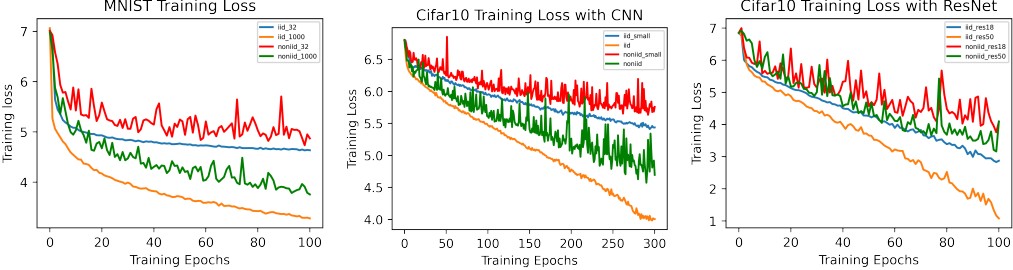

Figure 5: Log-scale loss.

