# OpenReview forum: "FedAvg Converges to Zero Training Loss Linearly: The Power of Overparameterized Multi-Layer Neural Networks"
_ICLR.cc/2023/Conference — Submitted to ICLR 2023_

### Official Review · Reviewer_vggk · 2022-10-18

**Confidence:** 3
**Correctness:** 4
**Technical Novelty And Significance:** 2
**Empirical Novelty And Significance:** Not applicable
**Recommendation:** 3

**Clarity, Quality, Novelty And Reproducibility:**

This paper has high quality regarding its writing: it clearly defines every symbol, which is not easy especially when notations are very complicated; and there is an easy-to-understand proof sketch. But the contribution of this work might be limited as described in the weakness section.

**Strength And Weaknesses:**

***Strength***
* It is the first convergence analysis of FedAvg over multi-layer neural network. And it is interesting to see it achieves linear convergence rate without additional requirements on data distribution.

***Weaknesses***
* I feel that the linear convergence of overparameterized model in the federated learning scenario is trivial. See the following two references. The data heterogeneity is characterized by the gap between global minimum and average of client minimum [Li2019] or the gradient noise at the global minimizer [Khaled2019]. For overparameterized model, both of those value are zero. It generally says that for overparameterized model there is no "data heterogeneity". I would be expecting to hear how data heterogeneity is defined in this paper and how it is different from those existing definitions.

*[Li2019] Li, X., Huang, K., Yang, W., Wang, S., & Zhang, Z. (2019, September). On the Convergence of FedAvg on Non-IID Data. In International Conference on Learning Representations.*

*[Khaled2019] Khaled, A., Mishchenko, K., & Richtárik, P. (2019). First analysis of local gd on heterogeneous data. arXiv preprint arXiv:1909.04715.*


* I agree that Theorem 1 indicates that FedAvg-SG attains a linear rate. But according to Remark 8 in the appendix, it appears that the choice of learning rate is very sensitive, it must be of the order $\Theta(\epsilon^2)$. If it is smaller, for example, $\eta =  \Theta(\epsilon^3)$, then the rate cannot be linear; and if $\eta =  \Theta(\epsilon)$, the convergence is not guaranteed. I don't think this result is satisfactory. For other research on overparameterized models, the following two references [Du2018] and [Ma2018] show the linear convergence rate can be achieved with a step size independent w.r.t. $\epsilon$. It is not clearly clarified in this paper why there is such uncommon constraints on the learning rate.

*[Du2018] Du, S. S., Zhai, X., Poczos, B., & Singh, A. (2018). Gradient descent provably optimizes over-parameterized neural networks. arXiv preprint arXiv:1810.02054.*

*[Ma2018] Ma, S., Bassily, R., & Belkin, M. (2018, July). The power of interpolation: Understanding the effectiveness of SGD in modern over-parametrized learning. In International Conference on Machine Learning (pp. 3325-3334). PMLR.*


===================================

Updates after Rebuttal:

I have read other reviews and the author's responses. I still tend to reject this paper. (1) As stated by the author in the rebuttal, this work uses a different definition of the data heterogeneity from [Li2019] and [Khaled2019]. So I don't think this result solves the standard heterogeneity issue in the federated learning. (2) Also, compared to prior over-parameterized model studies, this paper has non-standard constraints on the learning rate. I agree that multiple local updates will make -level learning rate invalid but it makes this paper meaningless: I can always run one-step local update at each edge device with a constant learning rate then applying [Du2018] and [Ma2018] to standard SGD with mini-batches; it also has a linear convergence rate. (3) Besides both above, I agree with Reviewer Er5f: there exists solid theoretical results showing that the minibatch SGD is always not worse than the federated averaging approach, so the motivation of sololy studying FedAvg is weak.

**Summary Of The Paper:**

This paper proposes an interesting solution for overcoming the data heterogeneity of FedAvg algorithm. The theoretical result indicates that for a specific structure of neural network, FedAvg can achieve nearly zero loss at a linear convergence rate without making any additional assumptions on data distribution. The empirical result further validates this result.

**Summary Of The Review:**

This paper fills in the blank of applying overparameterized model to FedAvg algorithms and achieves the expected result: linear convergence rate. But I have two major concerns: (1) the unclear definition of data heterogeneity; this paper may fall into an existing scoop of federated learning research. (2) the uncommon constraint on the step size; the theoretical result doesn't match some existing overparameterized models research.  From these two reasons, I tend to reject.

---

> ### Author Response · Authors · 2022-11-19
> **Response to Reviewer vggk**
>
> Thank you for the valuable comments and suggestions. We will address the weakness one by one.
>
> Weakness1[Novelty]: Thank you for your comment. First, we need to clarify that we analyze the convergence performance under no data homogeneity assumption. The data homogeneity we consider has the following formulation:
> \begin{equation}
> \|\nabla f_k(\theta)-\nabla f(\theta)\|\leqslant\delta,\;\forall \theta.
> \end{equation}
> The above formulation means the local gradient can be bounded by global gradient.First,
> we have checked the two data heterogeneity definition in the mentioned papers, which are different from our definition of data homogeneity. Second, we would like to clarify that the overparameterized model can't trivially satisfy the homogeneity conditon. This is because: (1) It is not clear if the global gradient is bounded in the domain. (2) The relationship between local gradient and global gradient is unclear.
>
> Weakness2[Choice of Stepsize]: From Theorem 1, our choice of stepsize $\eta\sim \mathcal{O}(\epsilon^2)$, which can guarantee the linear convergence of the loss function. We agree that the choice of $\eta$ is sensitive, but the strict restriction on stepsize $\eta$ is due to the distributed setting and stochastic algorithm we consider, which is more diffcicult than the case mentioned in comment ( [Du2018] and [Ma2018]). If we don't choose stepsize dependent on the chosen $\epsilon$, it is not possible to achieve arbitrarily small error.
> The difficulty lies in two aspects: (1) The stochastic algorithm makes the boundness of the gradient unclear. We do not know how large the scale of stochastic gradient is, so we have to choose small enough stepsize to guarantee the decrease after each update. (2) In distributed setting, the optimization is more challenging due to the difference between local gradient and global gradient. Thus, there is stronger restriction on the choice of stepsize.

---

> > ### Comment · Reviewer_vggk · 2022-12-09
> > **Thanks for the response**
> >
> > Thanks for the response. I have read other reviews and the responses. My concerns are not completely solved and I have updated it to the end of my original review. So I will keep my current score.

---

### Official Review · Reviewer_oX5J · 2022-10-21

**Confidence:** 4
**Correctness:** 4
**Technical Novelty And Significance:** 2
**Empirical Novelty And Significance:** 2
**Recommendation:** 3

**Clarity, Quality, Novelty And Reproducibility:**

The paper should provide a more clear discussion on what is the technical contribution compared with previous works, Nguyen and Mondelli (2020). Currently, I have the impression that many contents in this paper highly overlap that of Nguyen and Mondelli (2020), including the network architecture, the initialization, and part of the proof.

The key weakness of the FedAvg setting is that the learning rate is set so that more local steps make the first-order approximation easier rather than harder. Thus the key difficulty of analyzing FedAvg, namely the averaging part, is circumvented. I think to make the analysis meaningful or significant enough, at least the total update $\eta \cdot r$ should be of the order constant.

Along with Nguyen and Mondelli (2020), the significance or novelty of these results is questionable.


**Strength And Weaknesses:**

Strength:
1. A global and linear convergence result for FedAvg. In this sense, the result is new and interesting.

Weakness:
1. Compared with the result for centralized training (Nguyen and Mondelli, 2020), the technical contribution of this paper is questionable. The network architecture, the initialization, as well as most of the proof, are from Nguyen and Mondelli (2020). While in Remark 5, the technical novelty compared with it is discussed, either extending to federated training or dealing with stochastic gradient is not significant or novel enough.

2. The analysis of federated training is likely to be superficial or unrealistic in that, the total update length within each communication round, decreases as the local training steps increases. In particular, the step size $\eta \asymp A^{-r}$ ($A > 1$) where $r$ is the number of local steps so in total, $\eta r$ decreases exponentially. For one thing, this is never the real world case. Usually, each local client will train for several steps with constant step size and then aggregate on the server side. More locally training steps makes the averaging step harder, while in this paper's configuration, more local steps mean the total change of the parameter becomes even smaller, thus averaging also become easier. This lead to the theoretical aspect: this kind of configuration makes the total local updates within $r$ steps so small, that the first-order approximation is precise enough. This makes the $r$-step update FedAvg no harder than the $1$-step update FedSGD. It is then trivial to show that FedSGD indeed can handle arbitrary heterogeneity in the clients.

3. The presentation needs further polishing. I found several typos and notation inconsistencies in the equations.

**Summary Of The Paper:**

This paper proves under certain assumptions, FedAvg can converge linearly to the optimal solution even under heterogeneous clients. The convergence relies on specific network architectures and parameter initialization.

**Summary Of The Review:**

Due to the lack of technical novelty, I recommend rejection.

---

> ### Author Response · Authors · 2022-11-19
> **Response to Reviewer Ox5j**
>
> Thank you for the valuable comments and suggestions. We will address the weakness one by one.
>
> Weakness:
> 1.Compared with the result for centralized training (Nguyen and Mondelli, 2020), the technical contribution of this paper is questionable. The network architecture, the initialization, as well as most of the proof, are from Nguyen and Mondelli (2020). While in Remark 5, the technical novelty compared with it is discussed, either extending to federated training or dealing with stochastic gradient is not significant or novel enough.
>
> Reply: Thank you for your comment. As we have discussed in the paper, there are several **key differences** between  Nguyen and Mondelli (2020) and our work. Let us emphasize them again here: (1) The first key technical novelty of our analysis is to deal with the local updates at each client. Note that, because the clients can conduct local updates, and during local updates there is no communication with the server, so there is no guarantee that the overall objective will always decrease. This is in contrast to the original analysis in (Nguyen \& Mondelli, 2020), which has been focused on the centralized setting, where  every update reduces the objective. Therefore, instead of showing that the FedAvg algorithm always decreases the objective, we show that when the stepsizes are chosen properly, it is possible that the objective increases between two local updates, but the expected objective decreases between every consecutive communication round.
> (2) The second key technical novelty of our analysis comes from the use of stochastic gradients for local updates. A key step in our analysis is to characterize the relationship between the stochastic and full gradient updates. Specifically, in Lemma 1, we prove that the norm of stochastic gradients can be upper bounded by the norm of full gradients. This part of analysis is absent from (Nguyen \& Mondelli, 2020) since no stochastic algorithm has been considered there.
>
> 2. Thanks for your concern. Let us clarify our choice of stepsize. Regarding the exponential dependecy on local step number $r$, we want to clarify that this is standard, and is reasonbale when $r$ is a fixed number.
> In our work, similar to rest of FedAvg works the number of local updates $r$ is treated as a fixed constant. Moreover, note that the dependence on the factor $2^r$ is tackled by the step-size choice which is chosen inversely proportional to $2^r$. We note this this is standard as the other works on federated learning where the step-size diminishes with the number of local updates, for example see [Theorem 1] in [R1]. Also, to address reviewers' comment, we have made some minor changes to our proof to make the dependency on $r$ clear. Moreover, note that the initialization conditions (equation (69),(70)) ensure that $r$ cannot be arbitrarily large which is intuitive since choosing a very large $r$ might lead to client drift because of data heterogeneity. Thus, it is reasonable to have some exponential dependency on $r$, especially, when $r$ is a constant.
>
> 3. We have modified the orignial version to fix the typos.
>
> [R1] S. P. Karimireddy, S. Kale, M. Mohri, S. Reddi, S. Stich, and A. T. Suresh. Scaffold: Stochastic controlled averaging for federated learning. In International Conference on Machine Learning,pages 5132–5143. PMLR, 2020.10

---

> > ### Comment · Reviewer_oX5J · 2022-12-08
> > **Thanks for the response**
> >
> > Thanks for the response. I have read other reviews and the responses.
> >
> > I feel that the following argument is not so convincing:
> >
> > > The first key technical novelty of our analysis is to deal with the local updates at each client. Note that, because the clients can conduct local updates, and during local updates, there is no communication with the server, so there is no guarantee that the overall objective will always decrease.
> >
> > The clients indeed conduct local updates, and there is no communication with the server. But your step size is set very small so that local SGD will make the local loss decrease. And since the update is small, the server average will be almost equivalent to taking one gradient step of the global loss (sum of each client's loss).  I don't see anything special that makes it difficult to "guarantee that the overall objective will always decrease".

---

### Official Review · Reviewer_Er5f · 2022-10-21

**Confidence:** 4
**Correctness:** 3
**Technical Novelty And Significance:** 2
**Empirical Novelty And Significance:** 2
**Recommendation:** 3

**Clarity, Quality, Novelty And Reproducibility:**

The presentation of the current work needs to be improved:
1. In equations (2) and (3), what is the dimension of $W_l$ and $\theta$? In addition, what do you mean by $f_{L,k}(\theta)$? Do you mean is a function of just the output is dependent on $\theta$? Why do you have $F$ norm in equation (3)?
2. In definition 1, change $x$ to $\theta$.
3. Regarding Assumption 2, is the requirement for each client's model or the stacked model?
4. Why the condition in (7) depends on both $\theta_1$ and $\theta_2$ instead of just $\theta$?
5. There are lots of works that also make no assumptions on the data heterogeneity, e.g., Karimireddy 2020b, the authors should also discuss it.

**Strength And Weaknesses:**

The strength of the paper:
1. It shows a linear convergence rate of FedAvg under certain conditions.
2. Experiments validate the effectiveness of theoretical guarantees.

The Weaknesses of the paper:
1. The presentation is not clear. There are lots of unclear definitions, which makes the paper hard to follow. (see detailed comments in next section)
2. Assumption 3 seems to be a strong assumption on the activation functions. What are specific examples of this kind of activation function?
3. It is unclear whether FedAvg is better than the naive mini-batch SGD baseline in this setting. For example, if we do not perform local updates and instead compute $r$ stochastic gradients at the same point and send the mini-batch gradients to the server and then update the model, whether FedAvg can outperform this mini-batch SGD baseline? According to Lemma 1, such a procedure can reduce $\rho$, and it will also avoid the error introduced by local updates as in Lemma 2.
4. The paper only shows the convergence guarantee of FedAvg. Therefore, it is unclear the generalization performance of FedAvg using overparameterized multi-layer neural networks. Note that most of the existing works consider a stochastic problem, and thus their convergence guarantees of FedAvg can serve as the generalization guarantees.

**Summary Of The Paper:**

This paper studies the problem of federated learning when optimizing with overparameterized multi-layer neural networks. More specifically, the author shows that with a specialized initialization, FedAvg has a linear convergence rate when solving the federated learning problem with overparameterized multi-layer neural networks.

**Summary Of The Review:**

The problem, i.e., how the overparameterization will affect the performance of FedAvg, considered in this paper seems to be interesting. However, the results provided in the current paper are not strong enough. The main question I have is whether the FedAvg can outperform the naive mini-batch baseline in this setting. In addition, what is the generalization guarantee of the proposed method?

---

> ### Author Response · Authors · 2022-11-19
> **Response to Reviewer Er5f**
>
> Thank you for the valuable comments and suggestions. We will address the weakness one by one.
>
> Weakness:
> 2. Assumption 3 seems to be a strong assumption on the activation functions. What are specific examples of this kind of activation function?
>
> Reply: The Assumption 3 is strong, while the assumption is made to make analysis complete. Although restrictive assumptions are made in the theoretical analysis, these conditions are made to make the analysis more convenient. There is concrete example of activation that satisfies Assumption 3. See equation (2) and Fig. 4 in [R1], which is a family of parameterized ReLU functions, smoothened by a Gaussian kernel.
>
>
> [R1] Nguyen, Quynh N., and Marco Mondelli. "Global convergence of deep networks with one wide layer followed by pyramidal topology." Advances in Neural Information Processing Systems 33 (2020): 11961-11972.
>
> 3.It is unclear whether FedAvg is better than the naive mini-batch SGD baseline in this setting. For example, if we do not perform local updates and instead compute r stochastic gradients at the same point and send the mini-batch gradients to the server and then update the model, whether FedAvg can outperform this mini-batch SGD baseline? According to Lemma 1, such a procedure can reduce $\rho$ and it will also avoid the error introduced by local updates as in Lemma 2.
>
> Reply: We need to point out that our result **does not** imply FedAvg can outperform mini-batch SGD. In fact, it is never our intentention to compare FedAvg with mini-batch SGD. Our focus in entirely on analyzing the FedAvg algorithm itslef. We are not sure why the reviewer thinks that we have made such a comparison in the paper; It would be great if this comment could be more specific. Second, if we do not perform local updates and computations, and compute r stochastic gradients at the same time the parameter $\rho$ in Lemma 1 will be smaller. Smaller $\rho$ means faster convergence. To be specific, please see the choice of stepsize $\eta$ in equation (153) on Page 26.
>
>
> 4.The paper only shows the convergence guarantee of FedAvg. Therefore, it is unclear the generalization performance of FedAvg using overparameterized multi-layer neural networks. Note that most of the existing works consider a stochastic problem, and thus their convergence guarantees of FedAvg can serve as the generalization guarantees.
>
> Reply: Thank you for your suggestion. In this paper, we focus on the optimization performance of FedAvg with overparameterized networks, and we never claimed, or implied that we study generalization behavior of the algorithm. In fact, the majority of federated learning related works that publish in major machine learning conferences such as ICML, NeurIPS, ICLR  **only focus on the optimization** behavior of FedAvg, or its extentions; please see [R1,R2,R3]. In fact, it is widely accepted in the community that it is important to unerstand the optimization performance of this class of problem. While we agree that generalization performance is important,  we we do not see any issue to focus on the optimization sisde of the story.  After all, for a 9-page conference paper, we have to have our focus and cannot study everything about an algorithm. We hope that the reviewer will agree with our point.
>
>
> Clarity, Quality, Novelty And Reproducibility:
>
> 1.Reply: As we have discussed in equation (1), $W_{l}$ has the same dimension as $W_{l,k}$, which is  $\mathbb{R}^{n_{l-1}\times n_l}$. $\theta$ is the stacked parameters of all layers, which has dimension $\sum_{l=1}^{L}n_{l-1}\times n_{l}$.
>
> 2. We have modified it.
>
> 3. Assumption 2 is made for each client model. The server model has the same structure as client model, so this assumption also holds for the server model.
>
> 4. We have checked the original paper [R2] and modified it.
>
> 5. Thank you. We will add these works in the revised version.
>
> [R1] Khaled, Ahmed, Konstantin Mishchenko, and Peter Richtárik. "First analysis of local gd on heterogeneous data." arXiv preprint arXiv:1909.04715 (2019).
> [R2] Li, Xiang, et al. "On the convergence of fedavg on non-iid data." arXiv preprint arXiv:1907.02189 (2019).
> [R3] Acar, Durmus Alp Emre, et al. "Federated learning based on dynamic regularization." arXiv preprint arXiv:2111.04263 (2021).

---

### Official Review · Reviewer_3BVM · 2022-10-27

**Confidence:** 4
**Correctness:** 3
**Technical Novelty And Significance:** 3
**Empirical Novelty And Significance:** 2
**Recommendation:** 6

**Clarity, Quality, Novelty And Reproducibility:**

(Clarity) This work is well presented.

(Quality) High quality work, makes important contributions.

(Novelty) Novel.

(Reproducibility) Good.


===============================================================

Typo: In Remark 1, 'Assumption 2 is required to establish a PL like property' -> 'Assumption 3 is required to establish a PL like property'.

**Details Of Ethics Concerns:**

N/A.

**Strength And Weaknesses:**

Strength:

1. Given FedAvg is a widely used algorithm in FL and the interpolation regime is an interesting regime in FL. Built on the previous results on overparameterized models with pyramidal topology [NM2020], this paper makes important contributions to provide linear convergence rates of FedAvg in this setting.

2. The theoretical results are derived without assuming heterogeneity conditions.

Weaknesses:

1. The setup of experiments in Section 5 do not align with the theoretical results. I agree it is interesting to investigate the convergence performance of FedAvg under the settings in Sec 5. I would like to see the empirical performance of FedAvg with the specified initialization strategy for solving a problem where all the assumptions (network architecture and activation), which could better corroborate the theoretical results. Moreover, this could be helpful for examining the tightness of the convergence analysis (e.g., the theoretical linear convergence rate v.s. the actual linear convergence rate).

2. [minor] The non-iid setting considered in Sec 5 is not very challenging, it might be interesting to study the performance of FedAvg under more extreme data heterogeneity conditions, e.g., settings in Table III in [LDC+2021].

Questions:

1. The training loss is pretty large in Figure 2 and Figure 3, is the training loss in those plot the summed loss or averaged loss? Also, it might be helpful to visualize the training loss in log scale to study the linear convergence of FedAvg.


References

[NM2020] Global convergence of deep networks with one wide layer followed by pyramidal topology. Q. Nguyen and M. Mondelli. NeurIPS 2020.

[LDC+2021] Federated Learning on Non-IID Data Silos: An Experimental Study. Qinbin Li, Yiqun Diao, Quan Chen, Bingsheng He. https://arxiv.org/abs/2102.02079.

**Summary Of The Paper:**

This paper studies the convergence performance of FedAvg for training (pyramidal topology) overparameterized deep networks. Specifically, under certain assumptions, the authors design a special initialization strategy for FedAvg and prove linear convergence rate of FedAvg on training such overparameterized models without making assumptions on data heterogeneity. The authors also provide experimental results on MNIST and CIFAR10 under different data heterogeneous settings to study the effect of model size in FedAvg training.

**Summary Of The Review:**

This paper studies an important problem in FL optimization (convergence rate of FedAvg under overparameterized regime) and obtains interesting theoretical results, I would recommend acceptance.

---

> ### Author Response · Authors · 2022-11-19
> **Response to Reviewer 3BVM**
>
> Thank you for the valuable comments and suggestions. We will address the weakness one by one.
>
> 1.The setup of experiments in Section 5 do not align with the theoretical results. I agree it is interesting to investigate the convergence performance of FedAvg under the settings in Sec 5. I would like to see the empirical performance of FedAvg with the specified initialization strategy for solving a problem where all the assumptions (network architecture and activation), which could better corroborate the theoretical results. Moreover, this could be helpful for examining the tightness of the convergence analysis (e.g., the theoretical linear convergence rate v.s. the actual linear convergence rate).
>
> Reply: Thank you for your comments. We would like to clarify that, our experiments align with the theoretical setting. First, regarding the initialization strategy, we have implemented both random initialization and the initialization strategy that satisfies the condition in equation (23) and (24). Fig. 1 and Fig. 2 are based on the random initialization while Fig. 3 uses the special initialization strategy aligned with equation (23) and (24). Our results show that, both initializations lead to similar performance, so that the special initialization is not necessary in practice. Second, regarding the network architecture and activation, we would like to point out that the Assumptions 2-3 are only made for analysis. In experiment, it is not necessary to implement the method to satisfy these assumptions. To be specific, it is a common practice that restrictions and special assumptions are made (on either the network structure, or the optimization algorithm, or both) when analyzing the theoretical behavior of neural networks. For example, in [R1], the infinite width network is analyzed but wide network is used in simulation to approximate the infinite width; in [R2], the specific stepsize is required to achieve the convergence result, while in the experiment it is not verified if the stepsize satisfies the assumption. Third, we have made the plot of the logarithm of loss function versus the iteration number in supplemetary material (Fig 5)The result shows the decrease of loss function is linear versus the iteration number, which shows the tightness of our result.
>
> [R1] Jacot, Arthur, Franck Gabriel, and Clément Hongler. “Neural tangent kernel: Convergence and generalization in neural networks.” Advances in neural information processing systems 31 (2018).
> [R2] Li, Yuanzhi, and Yang Yuan. “Convergence analysis of two-layer neural networks with relu activation.” Advances in neural information processing systems 30 (2017).
>
> 2.[minor] The non-iid setting considered in Sec 5 is not very challenging, it might be interesting to study the performance of FedAvg under more extreme data heterogeneity conditions, e.g., settings in Table III in [LDC+2021].
>
> Reply: Thank you for your suggestion. We have checked the setting in the paper [LDC+2021]. This is an interesting setting which deserves to work on in the future. However, our non-iid setting is quite common in literature. For example, in [R3], [R4], same non-iid setting is used as our work.
>
> [R3] B. McMahan, E. Moore, D. Ramage, S. Hampson, and B. A. y Arcas. Communication-efficient learning of deep networks from decentralized data. In Artificial intelligence and statistics, pages 1273–1282. PMLR, 2017.
> [R4] Fedpd: A federated learning framework with adaptivity to non-iid data. IEEE Transactions on Signal Processing, pages 1–1, 2021. doi: 10.1109/TSP.2021.3115952.
>
> Question1: The training loss is pretty large in Figure 2 and Figure 3, is the training loss in those plot the summed loss or averaged loss? Also, it might be helpful to visualize the training loss in log scale to study the linear convergence of FedAvg.
>
> Reply: It is the sum of the loss, we have uploaded some plots of training loss in log scale.

---

### Decision · Program_Chairs · 2023-01-20

**Decision:**

Reject

**Justification For Why Not Higher Score:**

For the weakness points mentioned above.

**Justification For Why Not Lower Score:**

N/A

**Metareview: Summary, Strengths And Weaknesses:**

- Summary:

This paper studies the convergence performance of FedAvg for training (pyramidal topology) overparameterized deep networks. Specifically, under certain assumptions, the authors design a special initialization strategy for FedAvg and prove linear convergence rate of FedAvg on training such overparameterized models without making assumptions on data heterogeneity. The authors also provide experimental results on MNIST and CIFAR10 under different data heterogeneous settings to study the effect of model size in FedAvg training.

- Strengths
Findings of the paper include:
1. The paper makes contributions to provide linear convergence rates of FedAvg in the specific case considered in the paper (assuming heterogeneity assumptions).

- Weaknesses
1. Despite the response, Assumption 3 seems to be a strong assumption on the activation functions. The authors should consider dropping such an assumption to follow what the majority of literature assumes.
2. Compared to previous results, extensions to stochastic cases vs full gradient steps might not be sufficient for acceptance, unless a new theoretical technique / proof difficulties emerge. E.g., it would be better if the authors were explaining better the difficulty of applying these changes and what new comes out of this proof technique.
3. The use of heterogeneity in the paper is questioned how is different than existing works and how it adds to the difficulty of the task.

- What would be missing:
Overall, the authors could/should consider:
1. Dropping/"smoothing" proof-enabling assumptions
2. Make explicit statements + contributions how this theoretical extension is significantly different/more challenging than existing work.
3. This extends to the case of heterogeneity in the problem definition: reviewers still believe that clarification is needed whether this is significantly different case than existing works